



# Evaluating the Twentieth Century Reanalysis Version 3 with synoptic typing and East Antarctic ice core accumulation

Max T. Nilssen[1], Danielle G. Udy[1,2,3], and Tessa R. Vance[2]

[1]Institute for Marine & Antarctic Studies, University of Tasmania, Battery Point 7004, Australia
[2]Australian Antarctic Program Partnership, Institute for Marine & Antarctic Studies, University of Tasmania, Battery Point 7004, Australia
[3]Australian Centre for Excellence in Antarctic Science, Institute for Marine & Antarctic Studies, University of Tasmania, Battery Point 7004, Australia

**Correspondence:** Tessa Vance (tessa.vance@utas.edu.au)

**Abstract.** Weather systems in the southern Indian Ocean influence East Antarctic precipitation variability and surface mass balance. However, long term variability in synoptic-scale weather systems in this region is not well understood due to short instrumental records that are mostly limited to the satellite era (post 1979). Ice core records from coastal East Antarctica suggest significant decadal variability in snowfall accumulation, indicating that data from the satellite era alone is not enough to characterise climate variability in the high southern latitudes. It is therefore challenging to contextualise recent precipitation trends and extremes in relation to climate change in this area. We used synoptic typing of daily 500 hPa geopotential height anomalies and the Law Dome ice core (East Antarctica) annual snowfall accumulation record to investigate whether the Twentieth Century Reanalysis project can represent the synoptic conditions associated with increased precipitation at Law Dome prior to the satellite era. Twelve synoptic types were identified using self-organising maps based on their dominant pressure anomaly patterns over the southern Indian Ocean, with four types associated with above average daily precipitation at Law Dome. Our results indicate that the Twentieth Century Reanalysis project can reliably represent the meridional synoptic conditions associated with increased precipitation at Law Dome from 1948, aligning with the assimilation of consistent surface pressure data from weather stations in the southern Indian Ocean. This extends the time period available to contextualise recent trends and extremes in precipitation and synoptic weather conditions by up to three decades beyond the satellite era. These results will help contextualise East Antarctic surface mass balance variability prior to the satellite era, with implications for improved understanding of the largest source of potential sea level rise.

## 1 Introduction

Understanding the long-term variability in Antarctic surface mass balance is important to the global climate because the Antarctic ice sheet is the largest source of potential sea level rise (Fretwell et al., 2013). Precipitation is the dominant source of surface



mass balance variability (Lenaerts et al., 2019), however the seasonal and interannual variability of precipitation is less well understood in East Antarctica (Wille et al., 2021). Weather systems in the southern Indian Ocean, such as extratropical cyclones, fronts, and anticyclonic blocking influence precipitation variability in coastal East Antarctica (Catto et al., 2015; Uotila et al., 2011). These weather systems have changed in frequency over the satellite era in relation to modes of climate variability, such as the Southern Annular Mode (SAM) (Udy et al., 2021). However, the high latitudes of the Southern Hemisphere, and in particular the Indian Ocean sector, are data sparse due to the remote location and lack of long-term observational records. It is therefore challenging to contextualise recent precipitation trends and extremes in relation to climate change in this region, highlighted by a recent extreme atmospheric river event in March 2022 (Jones et al., 2016; Wille et al., 2021, 2024a, b).

Climate reanalyses that span the twentieth century can be used to help bridge this gap in understanding of past variability prior to the satellite era, and include the Twentieth Century Reanalysis (20CR, Slivinski et al., 2019; Compo et al., 2011), the European Centre for Medium-Range Weather Forecasts (ECMWF) Twentieth Century Reanalysis (ERA-20C, Poli et al., 2016), and the Coupled ECMWF Reanalysis of the Twentieth Century (CERA-20C, Laloyaux et al., 2018). All three reanalysis products assimilate surface pressure observations, while ERA-20C and CERA-20C also assimilate marine surface wind observations, and CERA-20C additionally assimilates ocean temperature and salinity. These reanalyses are generally considered to perform poorly in the Southern Ocean and Antarctic regions early in the twentieth century, due to the lack of observations. It has therefore been recommended to evaluate reanalysis model output such as daily precipitation against independent datasets, including ice core records of annual snowfall accumulation, to verify their performance (Schneider and Fogt, 2018; Wang et al., 2017, 2020).

Synoptic typing of atmospheric pressure patterns may provide additional insights that are not possible by simply comparing reanalysis precipitation to ice core snowfall accumulation records, by providing an indication of the atmospheric dynamics that are associated with the surface weather that drives precipitation variability. It has been demonstrated that synoptic typing methods, using both self-organising maps (SOM) and k-means clustering, can be used to investigate weather and climate variability in the southern Indian Ocean (Pohl et al., 2021; Udy et al., 2021). SOMs use a neural network algorithm with unsupervised learning to determine generalised patterns in large datasets (Kohonen, 1990), for example atmospheric pressure (Udy et al., 2021; Gibson et al., 2017). SOMs have been shown to be a useful tool in analysing synoptic conditions and frequency in the Southern Hemisphere (Gibson et al., 2017; Hope et al., 2006; Hosking et al., 2017; Verdon-Kidd and Kiem, 2009; Udy et al., 2021; Reusch et al., 2005).

In this study, we used 20CR to extend a prior synoptic typing dataset (Udy et al., 2021) to explore the representation of synoptic conditions that are associated with increased snowfall in East Antarctica prior to the satellite era. 20CR was chosen for this study because it only assimilates surface pressure observations, therefore it is less vulnerable to the inhomogeneities that can arise when new data is added or observing systems change (Slivinski et al., 2021; Compo et al., 2011). Although ERA-20C often outperforms 20CR in more well observed regions, 20CR may have an advantage in more data sparse regions such as the mid- and high-latitudes of the Southern Hemisphere, due to different data assimilation schemes and different kinds of assimilated observations (Hamill and Snyder, 2000; Gillespie et al., 2021).





We compared the annual frequency of the synoptic types to annual 20CR precipitation and the Law Dome ice core snowfall accumulation record (Roberts et al., 2015; Jong et al., 2022), to quantify the potential use of 20CR in the southern Indian Ocean prior to the satellite era. The study domain (30°-75°S, 40°-180°E), represents synoptic weather patterns that influence the coastal regions of the East Antarctic ice sheet. This region is the same as that used in the prior study of the satellite era (Udy et al., 2021) allowing comparison between the previous study and this temporally extended study.

## 2    Methods

### 2.1    Data Used

The Twentieth Century Reanalysis version 3 (20CRv3, Slivinski et al., 2019) is produced by the National Oceanographic and Atmospheric Administration, the University of Colorado Bulder's Cooperative Institute for Research in Environmental Sciences, and the U.S. Department of Energy (NOAA-CIRES-DOE). It assimilates only surface pressure observations from
the International Surface Pressure Databank version 4.7 (ISPDv4.7, Cram et al., 2015; Compo et al., 2019) using an Ensemble Kalman data assimilation system into an 80-member ensemble model. It covers 1836-2015, has a horizontal resolution of $1°×1°$, and a temporal resolution of 3 hours. All 20CRv3 data used here is the ensemble mean. 500 hPa geopotential height (z500) daily anomalies from 20CRv3 over the period 1900-2015 were calculated from the 1900-2015 climatological mean (Fig. 1).

The daily mean 3-hourly precipitation rate at 67°S, 113°E was used to represent the 20CRv3 precipitation at the Law Dome, Dome Summit South drill site (66.77°S, 112.81°E, 1370 m elevation). The 20CRv3 grid size at this latitude is approximately $111×43$ km$^2$. The precipitation rate was converted from its original units (kg m$^{-2}$ s$^{-1}$) into mean daily precipitation (mm day$^{-1}$). It is worth noting that Law Dome is a semi-independent ice cap which exhibits a strong orographic precipitation signal from east (high precipitation) to west (low precipitation) across the Dome (Pedro et al., 2011).

The annual snowfall accumulation record from the Law Dome DSS ice core was used in this study (Jong et al., 2022; Roberts et al., 2015). The DSS site receives relatively high annual snowfall compared to much of coastal East Antarctica, with a mean annual accumulation rate of 0.69 metres ice equivalent from frequent cyclonic incursions which produces seasonally varying annual layers in the ice core that can be accurately dated (Zhang et al., 2023; McMorrow et al., 2001; Mcmorrow et al., 2004; Plummer et al., 2012; Jong et al., 2022). Annual layers are identified using seasonal variations in water stable isotope ratios
$\delta^{18}$O and $\delta$D and trace chemistry, and validated against known volcanic eruptions (Roberts et al., 2015; Jong et al., 2022; Plummer et al., 2012).

### 2.2    Self-organising map inputs and evaluation

The SOM algorithm, from the Kohonen R package (Wehrens and Kruisselbrink, 2018), was used to identify regional synoptic patterns in the southern Indian Ocean from 1900 to 2015 using 20CRv3 daily 500 hPa geopotential height anomalies. The
code used in this study was adapted from Udy et al. (2021). The following grid parameters were selected: rectangular topol-





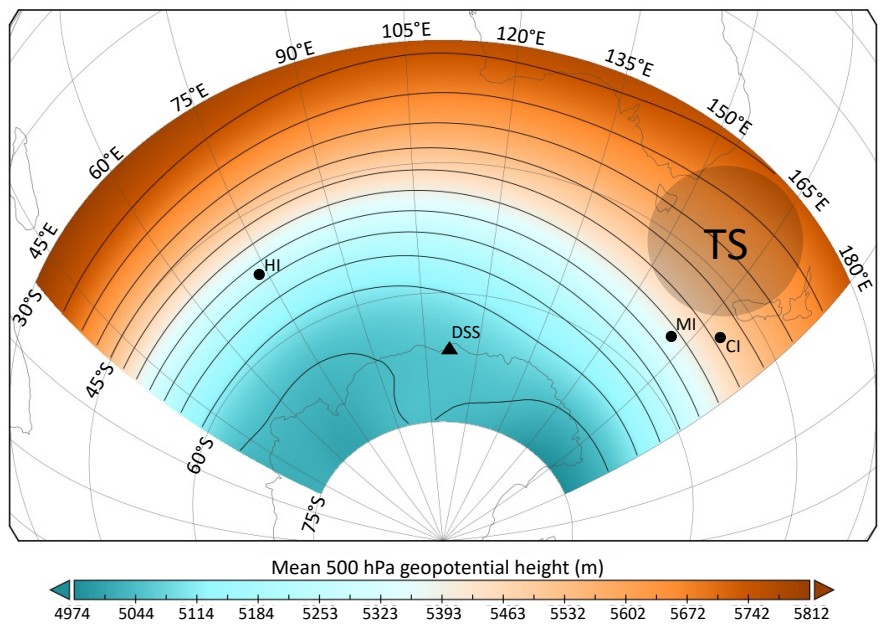

**Figure 1.** The study region, showing the 500 hPa geopotential height climatological mean (1900-2015). The locations of the Law Dome, Dome Summit South ice core site (DSS), Heard Island (HI), Macquarie Island (MI), Campbell Island (CI), and the Tasman Sea region (TS) are indicated. Map produced in Panoply with 20CRv3 data.

ogy, Gaussian neighborhood function, and Euclidean distance measurement. Training was carried out over 1000 iterations (improvement plateaued around 800 iterations), with the learning rate decreasing linearly from 0.05 to 0. The neighborhood radius parameter was consistent with Udy et al. (2021) who used a hybrid SOM-clustering approach (75% of training period SOM, 25% of training period K-means clustering) to improve the Euclidean distance and Pearson pattern correlation scores.

SOMs are similar to K-means clustering. The main difference between the two methods is that in the training stage for SOMs, the winning node and its surrounding nodes are updated at each iteration, while in K-means clustering, only the winning node is updated. The SOMs method is equivalent to K-means clustering when the radius parameter exactly equals 1 (Bação et al., 2005)). Further detailed descriptions of the SOM algorithm, and it has been applied to weather and climate applications, can be found in other studies (Hewitson and Crane, 2002; Sheridan and Lee, 2011; Verdon-Kidd and Kiem, 2009; Udy et al., 2021).

An important but subjective aspect of SOM analysis is choosing the number of nodes. When there are fewer nodes, the SOM output is more generalised. As the number of nodes increases, the nodes become more specific and a broader range of patterns can be discerned, but interpretability decreases. The ideal number of nodes will depend on the aims of a study. Here, we ultimately chose the configuration of 4 × 3 (12) nodes after testing the sensitivity of results to multiple node configurations including 3 × 3 (9), 4 × 3 (12), 3 × 5 (15) and 4 × 4 (16). The configuration of 4 × 3 (12) was chosen as it best represented

the range of zonal and meridional syntoptic types known to occur in the region (Udy et al., 2021; Pohl et al., 2021).





To evaluate how well the SOM output represented the actual daily synoptic patterns, the Pearson pattern correlation between each winning node and the corresponding z500 anomaly for each day was calculated. Note that Udy et al. (2021) calculated the Pearson pattern correlation between the winning nodes and the observed z500 field, rather than the z500 anomaly, which is less sensitive to small differences in the exact location of high/low anomalies. Therefore, the correlation scores here were expected to be lower than in Udy et al. (2021).

### 2.3 20CRv3 daily precipitation at Law Dome

To determine which synoptic types (i.e. SOM nodes) were associated with increased precipitation at Law Dome, the median daily 20CRv3 precipitation at the grid point closest to the Law Dome summit (67°S, 113°E) from each synoptic type was compared. To test the null hypothesis that the median precipitation associated with each synoptic type is equal, a Kruskal-Wallis rank-sum test was performed. A pairwise comparison using Wilcoxon rank sum test was then used to test which pairs of synoptic types had significantly different median precipitation.

The 90th and 99th percentile of 20CRv3 daily precipitation at Law Dome was calculated. Each day was classified as either "zero" (no precipitation), "normal" (<90th percentile), "high" (<99th, but >90th percentile) or "extreme" (>99th percentile). These statistics were used to evaluate the proportions of total annual precipitation that could be attributed to high and extreme precipitation days. This analysis was for the 20CRv3 precipitation only, and not for the synoptic types. We note that the terminology used for defining precipitation type days and percentile thresholds in this study is different to other studies (Jackson et al., 2023; Turner et al., 2019). For example, Turner et al. (2019) defined an extreme precipitation event as the top 10% of daily precipitation at a location, which is in contrast to our study which further defines the upper 10% of precipitation days as either high or extreme.

### 2.4 Correlation and regression analysis of snowfall accumulation

The 1900-2015 time period was split into sub periods based on changes in the number of observations assimilated into 20CRv3, based on the ISPDv4.7 dataset. This enabled the investigation of how correlations might change over time, with the goal of extending the utility of 20CRv3 beyond the satellite era. The three sub periods were:

1. 1979-2015 (modern satellite era, to allow for comparison with satellite era reanalyses)

2. 1957-2015 (assimilation of new data from the International Geophysical Year - 1957-1958)

3. 1948-2015 (assimilation of observations from Macquarie Island (Fig. 1) into ISPDv4.7 (Compo et al., 2019))

There were clear temporal trends in many variables, so linear detrending was performed. The dataset was split into sub periods first, and then each sub period was detrended. Following this, Pearson correlations were calculated between the 20CRv3 annual precipitation for the Law Dome grid cell and the DSS ice core annual accumulation record, to determine if 20CRv3 precipitation could accurately represent annual accumulation rates in the Law Dome region. Then, the Spearman correlation between the annual frequency of each synoptic type, the 20CRv3 total annual precipitation at Law Dome, and the DSS ice





core annual snowfall accumulation were calculated to evaluate whether the annual frequency of synoptic types correlated with annual snowfall accumulation at DSS (represented by either the 20CRv3 and/or the ice core dataset).

Linear models were generated to assess whether a linear combination of the annual frequency of the synoptic types could explain variability in the DSS ice core annual snowfall accumulation record. The variables were selected using the stepAIC function from the MASS (Modern Applied Statistics with S, Venables and Ripley, 2002) package in R. Models were generated for different time periods (see above). Plots showing the predicted DSS ice core annual accumulation were created using the predict function, with a 95% confidence interval around each point. All statistical analysis was carried out using RStudio Version 2023.03.0+386.

## 3 Results

### 3.1 Annual ice core accumulation compared to 20CRv3 precipitation

20CRv3 annual precipitation at the grid square containing Law Dome (67°S, 113°E) and the DSS ice core annual snowfall accumulation are significantly correlated over the full period examined in this study, 1900-2015 (Fig. 2 and Table 1). The correlation is strongest over the 1979-2015 period (r = 0.8, p<0.001). The 20CRv3 annual precipitation at Law Dome has a
positive trend over the 1900-2015 period (0.88 mm year$^{-1}$, p = 0.003), and in the 1900-1978 period (1.82 mm year$^{-1}$, p < 0.001), and a negative trend in the 1979-2015 period (-4.71 mm year$^{-1}$, p = 0.036). The DSS ice core annual snowfall accumulation record has no significant trend over the 1900-2015 period, but does have a negative trend over the 1979-2015 period (-4.91 mm year$^{-1}$, p = 0.034).

**Table 1.** Pearson correlation coefficient (r), and adjusted $R^2$, and p-value (two-sided Student's t test) for various time periods, showing correlation between 20CRv3 annual precipitation at Law Dome, and DSS ice core annual accumulation. The detrended values are shown first, with the non-detrended values shown in brackets. Note that the $R^2$ values are adjusted by accounting for the sample size.

| Time period | r value | Adjusted $R^2$ | p-value |
|---|---|---|---|
| 1900-2015 (116 yrs) | 0.548 (0.549) | 0.294 (0.296) | <0.001 |
| 1948-2015 (68 yrs) | 0.695 (0.695) | 0.475 (0.475) | <0.001 |
| 1957-2015 (59 yrs) | 0.743 (0.735) | 0.544 (0.532) | <0.001 |
| 1979-2015 (37 yrs) | 0.790 (0.815) | 0.613 (0.655) | <0.001 |

### 3.2 Contribution of high and extreme precipitation

The 90th percentile of 20CRv3 daily precipitation at Law Dome was 5.05 mm day$^{-1}$, and the 99th percentile was 13.4 mm day$^{-1}$. The amount of annual precipitation associated with high (90-99th percentile) and extreme (over 99th percentile) precipitation events increased over time (Fig. 2). The proportion of total annual precipitation from high precipitation days increased from the late 1940s, with a further increase from the mid-late 1950s. The proportion of precipitation that can be at-





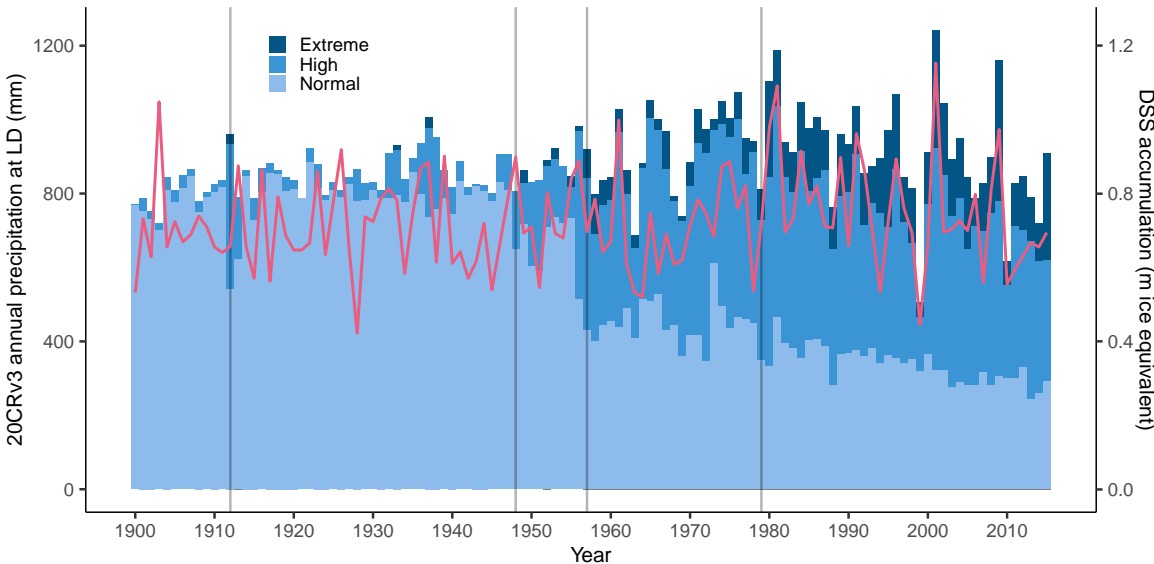

**Figure 2.** Annual precipitation variability comparison between 20CRv3 total annual precipitation (mm) at Law Dome (blue bars - shading indicates percentiles) and annual snowfall accumulation (miceequivalent) from the DSS ice core record (red line) over $1900 - 2015$. 20CRv3 total annual precipitation is split into 'types' based on percentiles; normal (<90[th] percentile, light blue), high (90-99[th] percentile, mid blue) and extreme (>99[th] percentile, dark blue) precipitation days. Vertical lines indicate the years 1912, 1948, 1957 and 1979 (see section 4.1)

tributed to extreme precipitation days increased from the mid-late 1950s, and a further increase from around 1980. The number
of zero precipitation days also increased from the late 1950s, with a continued positive trend until the 2000s. Prior to 1957, only two days were classified as zero precipitation days (one each in 1913 and 1952), and 13 days as extreme precipitation.

### 3.3 SOM output and evaluation

For initial testing, the SOM algorithm was performed on the 20CRv3 dataset for 1979-2015 with $3\times3$ (9) nodes (Fig. A1).
These nodes had the same broad patterns as the nine SOM nodes from ERA Interim (1979-2018) in Udy et al. (2021). This
confirmed that consistent results are achieved between the two reanalysis products for the common period of 1979-2015.

The final configuration of twelve nodes was chosen for this study based on their ability to represent the synoptic conditions
that result in snowfall at Law Dome. We determined that reducing the node number to 9 would not sufficiently capture the
range of possible weather patterns over the 116-year period. Using 15 and 16 nodes did show a broader range of synoptic
patterns, however the increase in the number of nodes from 12 did not substantially add information to help understand synoptic
variability in snowfall in the Law Dome region. The 12 nodes (synoptic types) are shown in Fig. 3.

Positive geopotential height anomalies are associated with high pressure ridges that extend poleward, while negative height
anomalies are associated with low pressure systems, such as extratropical cyclones and cold fronts. The structure and locations

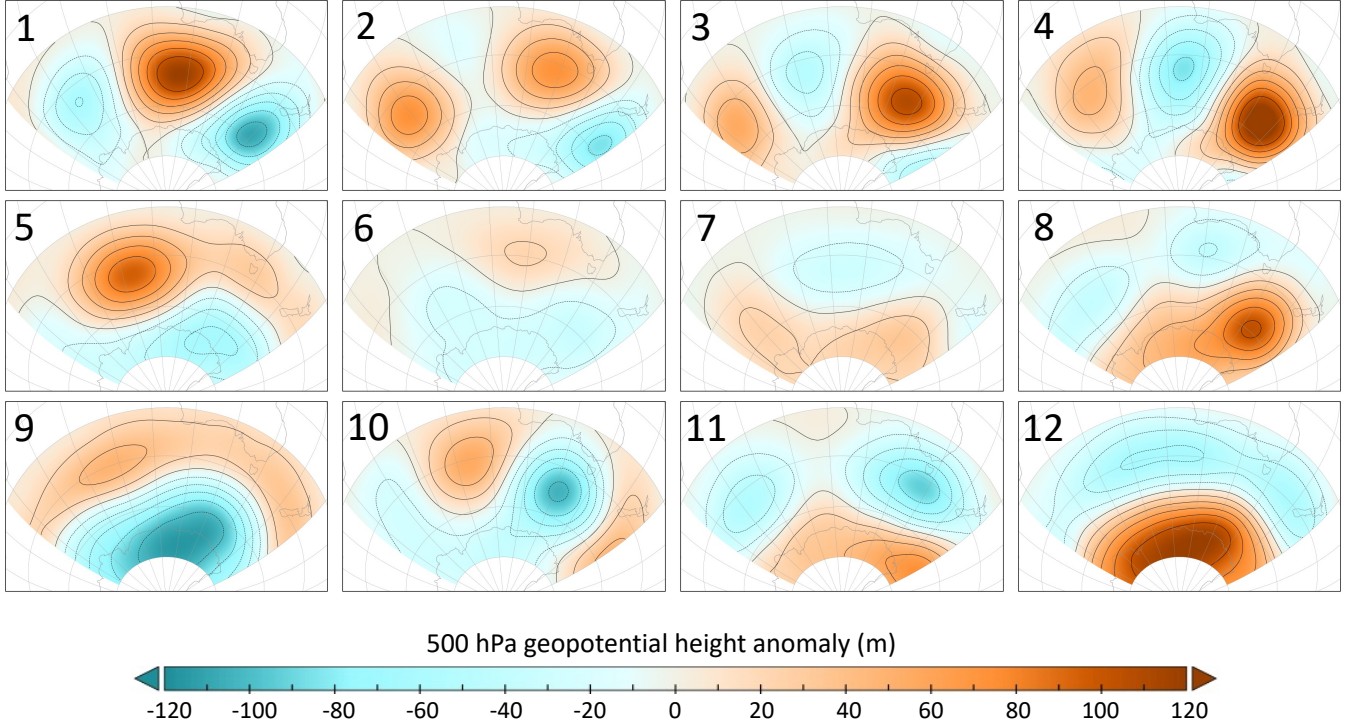

**Figure 3.** Self-organising map (SOM) output of 500 hPa geopotential height anomaly for each of the 12 SOM nodes (synoptic types). Positive height anomalies are shown in orange with solid contours, negative in teal with dotted contours. Maps produced in Panoply.

of the anomaly patterns are similar to Udy et al. (2021) as well as other studies in overlapping regions that use a combination of synoptic typing methods (Hope et al., 2006; Jiang et al., 2012; Pohl et al., 2021; Verdon-Kidd and Kiem, 2009)

The accuracy of the SOM output was quantified by calculating the Pearson pattern correlation (r) score between each winning node and the z500 anomaly at each daily time step and then analysing the distribution of the correlations (Fig. A3). The median r score across all nodes was 0.57, and 67% of the daily timesteps had r > 0.5. The performance metrics of most nodes displayed a normal distribution with a slight negative skew, except for nodes 6 and 7. These nodes were the least frequent, and displayed poor correlation scores, indicating that they represent a range of low frequency, but highly variable synoptic conditions.

## 3.4 Synoptic type descriptions

The 12 synoptic types are described in Table 2 and are grouped by their overall synoptic pattern. Similar to Udy et al. (2021) these broad groupings are defined as meridional (types 1, 3, 4 - display strong differences in anomalies east-west), zonal (types 9, 12 - display strong differences in anomalies north-south), mixed (types 2, 5, 8, 10, 11 - display a mixture of east-west and north-south differences in anomalies), and transitional (types 6, 7 - represent transitional states between other types).



**Table 2.** Structural groupings, anomaly descriptions and relative frequencies, and median r score of the 12 southern Indian Ocean synoptic types derived in this study. The frequency is over the entire study period (1900-2015). The median r score refers to the Pearson pattern correlation scores between the composite node and daily anomaly maps used to evaluate the performance of the SOM

| Synoptic type | Structure | Description | Overall frequency | Median r score |
|---|---|---|---|---|
| 11 | Mixed | Negative height anomalies centred at 48°S, 62°E and at 48°S, 148°E; positive anomaly over East Antarctica | 9.5% | 0.58 |
| 10 | Mixed | Weak positive height anomaly centred at 45°S, 95°E; negative anomaly centred at 52°S, 138°E | 8.6% | 0.54 |
| 8 | Mixed | Positive height anomaly in the Tasman Sea region, centred at 59°S, 156°E and spanning East Antarctica | 7.9% | 0.57 |
| 5 | Mixed | Positive height anomaly centred at 50°S, 95°E, which longitudinally spans the study area, with a negative anomaly over East Antarctica | 7.8% | 0.60 |
| 2 | Mixed | Positive height anomalies centred at 52°S, 60°E and 48°S, 132°E, with a weak negative anomaly at 58°S, 175°E | 8.5% | 0.58 |
| 6 | Transitional | Represents a wide variety of synoptic conditions, and transitions between more distinct synoptic types | 6.2% | 0.35 |
| 7 | Transitional | Represents a wide variety of synoptic conditions, and transitions between more distinct synoptic types | 5.8% | 0.41 |
| 9 | Zonal | Positive height anomaly in the midlatitude southern Indian Ocean and a negative height anomaly over East Antarctica. | 9.9% | 0.62 |
| 12 | Zonal | Negative height anomaly in the midlatitude southern Indian Ocean and a positive height anomaly over East Antarctica | 10% | 0.67 |
| 4 | Meridional | Positive height anomaly in the Tasman Sea region, centred at 55°S, 158°E, and a weak negative anomaly centred at 50°S, 113°E | 8.7% | 0.57 |
| 3 | Meridional | Positive height anomaly in the Tasman Sea region, centred at 53°S, 145°E, and a weak negative anomaly centred at 48°S, 95°E | 8.6% | 0.55 |
| 1 | Meridional | Positive height anomaly centred at 50°S, 115°E, with a strong negative height anomaly centred at 58°S, 168°E, and a weaker negative height anomaly centred at 50°S, 68°E | 8.4% | 0.59 |





## 3.5 Synoptic type inter-annual frequency and precipitation

The overall frequency of each synoptic type ranged from 5.8% to 10% (Table 2). The non-transitional synoptic types displayed frequencies between 7.8% to 10%, with the transitional synoptic types having slightly lower frequencies of 5.8% (type 7) and 6.2% (type 6). The zonal synoptic types displayed the highest interannual variation, with type 9 varying between 0% and 39%, and type 12 between 0.5% and 37%. The annual frequency of other types varied between 0.3% and 24%. The frequency of type 9 increased significantly over 1900-2015 (0.636 $\mathrm{days\ year^{-1}}$, p < 0.001). There was no significant trend in the frequency of type 12 for 1900-2015, but there was a negative trend for 1957-2015 (-0.459 $\mathrm{days\ year^{-1}}$, p < 0.001). The frequency of the transitional and mixed types decreased over time, especially types 6, 7, and 11, while the frequency of meridional types increased, especially types 1 and 4 (See Table B1 for trends in all variables). The transitions between synoptic types are summarised in Fig. A2.

Testing showed that at least one of the synoptic types had a median daily precipitation significantly different to the others (Kruskal-Wallis chi-squared = 6098, df = 11, p < 0.0001). Furthermore, a Wilcoxon rank sum test indicated that of the 66 synoptic types pairings, 59 had significantly different medians (Bonferroni adjustment, p < 0.05). Synoptic types 3, 4, 8 and 12 were associated with high precipitation at Law Dome (Fig. 5a, b). The annual frequency of each synoptic type was compared to the annual Law Dome precipitation from 20CRv3, as well as the annual snowfall accumulation rates from the DSS ice core record for different time periods (Table 3).

## 3.6 Linear model estimates of ice core annual accumulation from synoptic typing

Multiple linear regression models were generated to estimate the variability in DSS ice core snowfall accumulation explained by a linear combination of the annual frequency of synoptic types over 1948-2015, 1957-2015 and 1979-2015 (Fig. 6). The model for 1900-2015 was not significant and therefore not shown. Summaries of the linear model outputs can be found in Appendix C. The models capture the mean variability in DSS accumulation, but fail to represent the larger extremes (e.g. around 1960, 1980 and 2000).

## 4 Discussion

Our results indicate the 20CRv3 ensemble can represent the meridional synoptic conditions in the southern Indian Ocean that are required for high precipitation (90-99th percentile) at Law Dome back to 1948, and extreme (>99th percentile) and zero precipitation back to 1957 (Fig. 4b). This extends the period available to contextualise recent trends and extremes in precipitation and synoptic weather patterns by 2-3 decades prior to the satellite era.

### 4.1 Key observations improve precipitation and synoptic type variability in 20CRv3

Improvements in the ability of 20CRv3 to reflect meridional weather patterns (Fig. 3) in the southern Indian Ocean and to reproduce zero as well as high and extreme precipitation days at Law Dome (Fig. 4b) relies on the commencement of



**Figure 4.** Comparison between the number of observations, count of precipitation type days and the annual frequency of synoptic types over 1900-2015. a) The number of assimilated observations per year into ISPDv4.7, south of 50°S in the study area (50°-75°S, 40°-180°E). Note the log scale. b) The number of zero, high (90-99th percentile) and extreme (>99th percentile) precipitation days each year. c) Annual frequency of synoptic types, grouped by structure: mixed (purples), transitional (greys), zonal (greens), and meridional (pinks). Note the increasing frequency of type 9 (light green) and type 4 (light pink) from around 1975 and the short periods of increased frequency of type 4 in around 1912 and 1930s. Vertical lines indicate the years 1912, 1948, 1957 and 1979 (see section 4.1).





**Figure 5.** 20CRv3 daily precipitation associated with each synoptic type at Law Dome (67°S, 113°E), for 1900-2015. a) Count of precipitation condition days per synoptic type; zero (grey), normal (<90th percentile, light blue), high (90-99th percentile, medium blue) and extreme (>99th percentile, dark blue) precipitation days. Types 3, 4, 8 and 12 have a higher proportion of high and extreme precipitation days, while types 1 and 9 have a higher proportion of zero days. b) Box and whisker plots of daily precipitation (mm) by each synoptic type. The overall median daily precipitation (1.79 mm day$^{-1}$) is indicated by red dashed line , and the purple bar indicates the interquartile range (0.80 mm day$^{-1}$, 3.03 mm day$^{-1}$). Synoptic types 3, 4, 8 and 12 have higher median daily precipitation than the overall median.



**Figure 6.** The DSS ice core annual snowfall accumulation (red line), and the predicted DSS accumulation based on selected synoptic types (black line), with 95% confidence interval around each point (grey shading), for (a) 1979-2015, (b) 1957-2015, and (c) 1948-2015. The accumulation is predicted using the annual frequency of different combinations of synoptic types, e.g. the 1957-2015 model only uses the annual frequency of types 1, 9 and 11. Refer to Appendix C for more detail on the combination of synoptic types and linear model output.





**Table 3.** Linearly detrended Spearman correlation (r scores) and statistical significance (two-sided Student's t-test) between the annual frequency of each node, and the annual 20CRv3 precipitation at Law Dome, and the annual snow accumulation rate at DSS (ice core) for the four different time periods. Note the high precipitation synoptic types (3, 4, 8, 12), and the low precipitation type (1). Correlation significance is indicated as follows: > 99% bold type, > 95% normal type, < 95% italic type.

| Synoptic type | 1900-2015 | | 1948-2015 | | 1957-2015 | | 1979-2015 | |
| | Precip (20CR) | Accum (DSS) | Precip (20CR) | Accum (DSS) | Precip (20CR) | Accum (DSS) | Precip (20CR) | Accum (DSS) |
|---|---|---|---|---|---|---|---|---|
| 1 | **-0.28** | *-0.13* | **-0.33** | **-0.33** | **-0.37** | -0.33 | -0.34 | *-0.24* |
| 2 | *-0.07* | *0.00* | *0.21* | *0.22* | 0.27 | 0.26 | *0.28* | *-0.02* |
| 3 | **0.24** | *0.09* | *0.23* | *0.17* | 0.22 | 0.19 | 0.26 | 0.26 |
| 4 | **0.25** | *0.01* | *0.20* | *0.00* | 0.21 | 0.08 | 0.15 | *-0.03* |
| 5 | *-0.16* | *-0.13* | *-0.05* | *0.11* | *-0.01* | 0.18 | *-0.05* | 0.06 |
| 6 | *-0.01* | *0.00* | *-0.06* | *0.02* | *-0.06* | 0.05 | *-0.21* | *-0.17* |
| 7 | *-0.09* | *-0.15* | *-0.06* | *-0.02* | *0.01* | 0.04 | *0.01* | 0.05 |
| 8 | **0.28** | *0.11* | 0.31 | *0.20* | **0.35** | 0.18 | 0.35 | 0.35 |
| 9 | -0.23 | *-0.10* | -0.28 | *-0.18* | -0.33 | -0.20 | -0.36 | *-0.26* |
| 10 | *-0.18* | *0.03* | *-0.22* | *-0.07* | *-0.19* | -0.09 | *-0.04* | *0.10* |
| 11 | *-0.02* | *0.01* | *0.01* | *-0.07* | *-0.04* | -0.18 | *0.18* | *-0.05* |
| 12 | **0.34** | *0.15* | **0.33** | *0.13* | 0.32 | 0.20 | 0.27 | *0.18* |

atmospheric pressure observations from weather stations and ships in key locations across the southern Indian Ocean. There are several locations in the study area where observations that have been assimilated into ISPDv4.7 increase from the late 1940s (Compo et al., 2019). Observations from ships in the southern Indian Ocean increased from 1946 coinciding with the end of the second World War. The Australian National Antarctic Research Expeditions (ANARE) also established bases on Heard Island in 1947 and Macquarie Island in 1948 (Dodd, 2023). Observations from stations on the New Zealand mainland and Campbell Island (52.54°S, 169.14°E) commenced from 1949 (Compo et al., 2019). The International Geophysical Year (July 1957- December 1958) also saw a large increase in meteorological observations in East Antarctica (Wexler, 1956).

Prior to these key observations, variability in annual precipitation derived from the 20CRv3 is low compared to the later period, and does not align with variability in the Law Dome DSS ice core annual snowfall accumulation record (Fig. 2, Table 1). The 20CRv3 also displays a reduced ability to generate high precipitation days before the late 1940s, as well as zero and extreme precipitation days before the late 1950s (Fig. 4b). High/extreme precipitation days are known to account for more than 40% of the total annual precipitation across East Antarctica (Turner et al., 2019). In contrast, only 7% of total annual 20CRv3 precipitation at Law Dome came from high/extreme precipitation (>90[th]) events for 1900-1947, compared to 24% between 1948-1956, and 59% between 1957-2015. This increase in the number of high/extreme precipitation days explains the apparent positive trend in the 20CRv3 annual precipitation over the period 1900-2015 (Table B1). This positive trend is likely an artefact of increased observations in ISPDv4.7 as the Law Dome DSS annual snowfall accumulation record does not





indicate any trends over this time period (Table B1). Spurious trends related to observation density is a well studied problem in reanalyses (e.g., Bromwich and Fogt, 2004; Huai et al., 2019; Marshall and Harangozo, 2000; Thorne and Vose, 2010; Wang et al., 2016). It has been shown that reanalyses that span the twentieth century undergo a change in/around 1950 in the high latitudes of the Southern Hemisphere, including a reduction in standard error, a spurious drop in surface pressure (Schneider

and Fogt, 2018), and a jump in 20CR precipitation − evaporation (P−E) over the East Antarctic Ice Sheet (Wang et al., 2020).

Short periods of increased observations in 1912 and during the 1930s (Fig. 4a) align with increased frequencies of high precipitation days in 20CRv3 model output (Fig. 4b) and increased frequency of synoptic type 4, one of the four synoptic types associated with high/extreme precipitation at Law Dome (Fig. 5b). Sir Douglas Mawson's 1911-1914 Australasian Antarctic Expedition recorded a large number of surface pressure observations from Macquarie Island. There were also observations in

this period from coastal East Antarctica, attributed to the expeditions of Mawson and Scott (Mawson, 1914; Hesselberg, 1922). The increase in observations in the 1930s (Fig. 4a) is more difficult to assign to specific weather stations or expeditions.

Atmospheric pressure data from Macquarie Island (54.62°S, 158.86°E) is particularly important for distinguishing high/extreme precipitation at Law Dome, as it is sensitive to the strength and position of anticyclonic blocking in the Tasman Sea region. Anticyclonic blocking in the SW Pacific, and particularly in the Tasman Sea, are key atmospheric drivers of episodic

high precipitation events across East Antarctica (Scarchilli et al., 2011; Udy et al., 2022; Pohl et al., 2021). Synoptic types 4 and 8 both reflect Tasman Sea anticyclonic blocking and increased precipitation at Law Dome (Fig. 3, Fig. 5a, b).

### 4.2 Extending synoptic typing in the southern Indian Ocean beyond the satellite era

The twelve synoptic types in this study were compared to the nine synoptic types in Udy et al. (2021), which will be referred to as Udy SOM3, Udy SOM6, etc. Many (but not all) of the synoptic types have direct comparisons, due to the different number

of synoptic types, and the longer time period in this study compared to Udy et al. (2021). Synoptic types 3, 4, and 8 (this study), which were associated with high precipitation at Law Dome (Fig 5b), display strong positive height anomalies in the Tasman Sea region (Fig. 3) and are comparable to Udy SOM2/SOM6. These positive height anomalies represent anticyclonic blocking patterns that increase precipitation in East Antarctica (Scarchilli et al., 2011; Servettaz et al., 2020; Udy et al., 2021, 2022), and are often associated with atmospheric rivers (Pohl et al., 2021; Wille et al., 2021). The zonal anomaly patterns in synoptic

types 9 and 12 are consistent with the zonal symmetry expected with the Southern Annular Mode (SAM) (Rogers and Loon, 1982). Synoptic type 9 has a positive anomaly in the mid-latitudes, consistent with positive SAM (SAM+) (Fogt and Marshall, 2020). Synoptic type 12 has a negative anomaly in the mid-latitudes, consistent with a negative SAM (SAM-) pattern and Udy SOM3 (Udy et al., 2021; Fogt and Marshall, 2020). Type 12 is associated with high precipitation days (Fig. 5b), which is consistent with previous studies that have found that SAM- is associated with increased precipitation in the Law Dome region

(Marshall et al., 2017). The enhanced strength of polar easterlies and positive precipitation anomaly in Udy SOM3 suggests the precipitation associated with synoptic type 12 is predominantly orographic precipitation, from moist air uplifted across Law Dome (Udy et al., 2021, 2022). Consistent with the positive trend in SAM over the twentieth century (Abram et al., 2014; Arblaster and Meehl, 2006; Jones et al., 2016), the annual frequency of synoptic type 9 (SAM+) increased over 1900-2015, while the frequency of synoptic type 12 (SAM-) decreased from around 1960 (Fig. 4c). Understanding SAM variability and the





associated synoptic-scale weather conditions prior to the satellite era is important, given the relationship between SAM, ozone depletion / recovery and rising greenhouse gases (King et al., 2023; Arblaster et al., 2011). SAM variability is also known to influence the mid-latitude climate variability of the Southern Hemisphere (Fogt and Marshall, 2020), including fire activity in southeast Australia (Abram et al., 2021).

### 4.3 Relationships between the annual frequency of synoptic types and DSS ice core accumulation

Only a few significant correlations exist between the annual frequency of individual synoptic types and the DSS ice core annual accumulation record (Table 3). The annual frequency of synoptic type 1, which is a low/zero precipitation type (Fig. 5), was the most consistent type to correlate with both 20CRv3 annual precipitation and the DSS ice core annual accumulation over the time periods tested. The annual frequency of synoptic type 8 was significantly ($p < 0.05$) positively correlated over the time periods tested with 20CRv3 annual precipitation, but was not significantly correlated with the DSS ice core annual
accumulation, with the exception of the 1979-2015 period.

Despite synoptic type 3 having the highest median daily precipitation (2.96 mm day$^{-1}$ compared to the overall median of 1.79 mm day$^{-1}$), the annual frequency of synoptic type 3 was not significantly correlated with the DSS ice core annual accumulation in any of the time periods tested. In contrast, synoptic type 3 was significantly correlated with 20CRv3 annual precipitation for only the 1900-2015 period. One explanation for the lack of significant correlation over more time periods is that only a
subset of surface weather conditions represented by synoptic type 3 are associated with high precipitation at Law Dome. That is, the positive height anomaly in the Tasman Sea region in synoptic type 3 could represent a blocking anticyclone that brings snowfall to Law Dome on some days, but on other days, where the anomaly is slightly offset to the west or east, the associated precipitation instead falls to the west or east of Law Dome. It is also possible that snowfall is blown away by wind and therefore would not appear in the ice core record. As is common with ice core records, annual precipitation at Law Dome is greater than
the net ice accumulation preserved in the ice core record due to loss processes such as wind erosion and ablation (McMorrow et al., 2001; Zhang et al., 2023).

A linear combination of multiple synoptic types explained more variability in the DSS ice core annual accumulation record than any single synoptic type (Fig. 6). The multiple linear regression models explain 30% of the variability in the DSS ice core annual accumulation record for the 1979-2015 period (Fig. 6a), 24% for the 1957-2015 period (Fig. 6b), and 17% for the
1948-2015 period (Fig. 6c). Perhaps not surprisingly, over 1900-2015, the variability explained is not significant ($p > 0.05$). The inclusion of synoptic type 1 in all of the multiple linear regression models (refer to Appendix C), suggests that synoptic conditions associated with low/zero precipitation days are important for reflecting annual snowfall variability, and underline the significance of understanding the episodic nature of precipitation across East Antarctica that results from frequent transitions between zonal and meridional synoptic patterns.

These results indicate that 20CRv3 can represent the combination of synoptic conditions that are associated with precipitation variability at Law Dome after 1948. The variability explained by the linear models suggest that the assimilated observations after 1948, while limited, incorporate enough spatial observations to force the reanalysis model ensemble toward a representation of real synoptic variability, rather than simply the climatological mean. This means that longer term reanalyses could be



used in future synoptic scale studies in the southern Indian Ocean back to 1948, vastly increasing the ability of reanalyses to
investigate topics as diverse as decadal changes in surface mass balance, ice core climate proxy stability and long-term variability in low-to-high latitude teleconnections. Udy et al. (2021) found links between synoptic weather patterns in the southern Indian Ocean and modes of climate variability, including SAM, the El Niño-Southern Oscillation and the Indian Ocean Dipole over the satellite era. These links to modes of variability have also been independently detected and used to reconstruct past variability using East Antarctic ice cores (Vance et al., 2013, 2015, 2022; Crockart et al., 2021). The extension of the 'useable'
reanalysis period by three decades (to 1950s) using 20CRv3, will greatly assist in developing as well as evaluating the skill of reconstructions of climate variability and their representation in climate models.

## 5   Conclusions

Our results indicate that 20CRv3 can reliably represent synoptic conditions associated with increased precipitation at Law Dome from the late 1940s. The incorporation of atmospheric pressure data from the Tasman Sea region (Macquarie Island
in the mid-twentieth century) is particularly important for the 20CRv3 models' ability to reflect meridional synoptic patterns in the southern Indian Ocean, and realistically represent annual precipitation variability at Law Dome. The 20CRv3 model ability to simulate high precipitation days begins to consistently improve after 1948, before stabilising after 1957. This shift in the ability to simulate high precipitation beyond 1948 leads to a spurious positive trend in 20CRv3 annual precipitation over 1900-2015 that is not supported by the DSS ice core accumulation record, and should be considered if using 20CRv3
precipitation in palaeoclimate or climate change studies. Further improvements and extension to the 'useable' period of the 20CRv3 could be realised if additional observations from ongoing data rescue efforts are included, which would greatly bolster investigations of long-term climate variability and changes to surface mass balance in East Antarctica. Initiatives like the Atmospheric Circulation Reconstructions over the Earth (ACRE) digitisation of historical weather data (e.g. logbooks, journals, and land data from expeditions) (Allan et al., 2011; Brönnimann, 2022; Brönnimann et al., 2018) are critical to this endeavour,
as our results highlight how valuable even small amounts of atmospheric pressure data can be to shift the reanalysis geopotential height patterns from 'climatology' to meridional patterns associated with increased precipitation.

*Code and data availability.*  The synoptic typing dataset developed using 20CRv3 over 1900-2015 will be available upon acceptance of this manuscript. The R code for self-organising maps and precipitation analysis used in this study will be made available upon acceptance.

    The base SOM code is available:
https://github.com/dgudy91/2021_AAPP_ML_workshop/blob/main/AAPP_ML_workshop_SOM_tutorial_data_subset.ipynb

    Other datasets used in this study are available online in the following locations:

    20CRv3: https://psl.noaa.gov/data/gridded/data.20thC_ReanV3.html

    DSS ice core accumulation: https://data.aad.gov.au/metadata/records/fulldisplay/DSS_2k_data_compilation

    ISPDv4.7 observation counts: https://psl.noaa.gov/data/20CRv3_ISPD_obscounts/



**Appendix A: Self-organising maps**

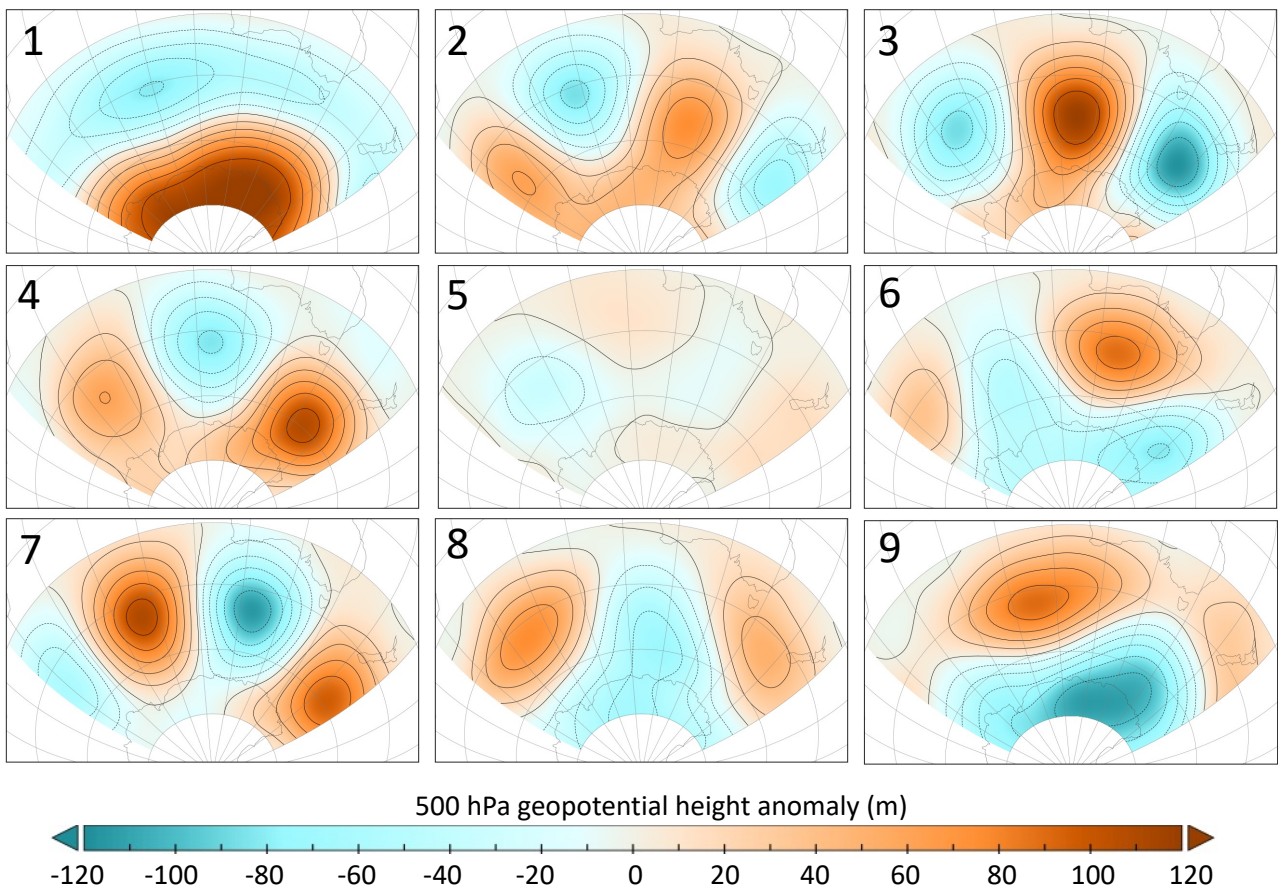

**Figure A1.** Self-organising map (SOM) output of 500 hPa geopotential height anomaly for each of the 9 SOM nodes. These were generated from daily anomalies from 20CRv3, 1979-2015. The nodes here display very similar patterns to the nodes in Udy et al. (2021), but they occur in a different order. For example, synoptic node 1 (this study) is equivalent to UdySOM3, and synoptic node 6 (this study) is equivalent to UdySOM4.





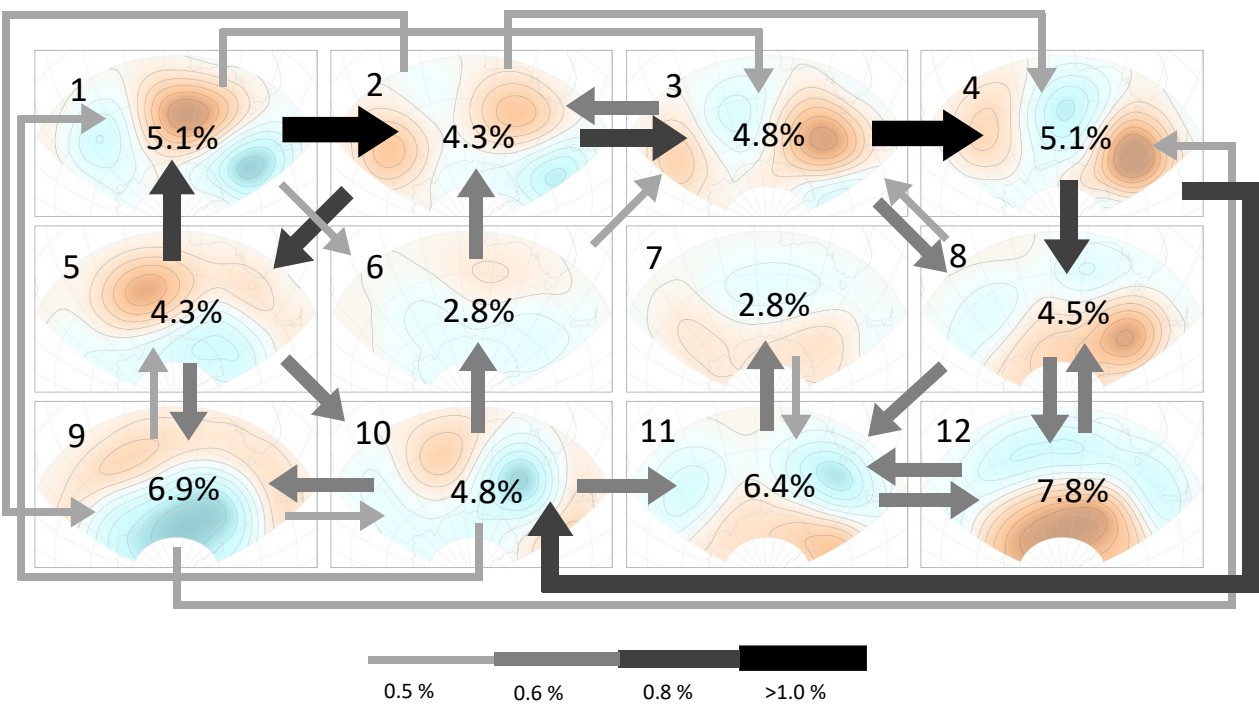

**Figure A2.** Frequency of daily persistent patterns and daily transitions between synoptic types over the study period. The percentage value shown on the composite map of each synoptic type represents the proportion of the study period where the same type persisted for two or more days. Daily persistence across all types accounted for 60% of the study period. Daily transitions greater than 0.5% of the study period ( 210 days) are shown, with thicker and darker arrows for higher percentages. The more frequent transitions are indicative of the expected eastward progression of weather systems in the study region (Udy et al., 2021). 32 transitions are shown, which make up 22% of the study period. Most types transitioned into every other type at least once, except for types 5, 9 and 10 which never transitioned into type 12, and types 11 and 12 which never transitioned into type 9.





**Figure A3.** Histograms of Pearson pattern correlation (r) score between the winning SOM node composite and the z500 anomaly of each daily time step assigned to that node.





## Appendix B: Trends

**Table B1.** Temporal trends of the 12 synoptic types, 20CRv3 annual precipitation at Law Dome, and DSS annual snow accumulation, for the 1900-2015 period. Note the weak but significant positive trend in 20CRv3 precipitation, with no trend in DSS accumulation.

| Variable | Slope coefficient | Units | Adjusted $R^2$ | p-value |
|---|---|---|---|---|
| Type 1 | 0.155 | days year$^{-1}$ | 0.156 | <0.001 |
| Type 2 | -0.085 | days year$^{-1}$ | 0.061 | <0.01 |
| Type 3 | 0.002 | days year$^{-1}$ | -0.009 | 0.9 |
| Type 4 | 0.275 | days year$^{-1}$ | 0.408 | <0.001 |
| Type 5 | -0.018 | days year$^{-1}$ | -0.006 | 0.6 |
| Type 6 | -0.201 | days year$^{-1}$ | 0.392 | <0.001 |
| Type 7 | -0.334 | days year$^{-1}$ | 0.542 | <0.001 |
| Type 8 | -0.143 | days year$^{-1}$ | 0.120 | <0.001 |
| Type 9 | 0.636 | days year$^{-1}$ | 0.487 | <0.001 |
| Type 10 | 0.083 | days year$^{-1}$ | 0.055 | <0.01 |
| Type 11 | -0.331 | days year$^{-1}$ | 0.372 | <0.001 |
| Type 12 | -0.039 | days year$^{-1}$ | -0.004 | 0.5 |
| 20CRv3 precip at Law Dome | 0.878 | mm year$^{-1}$ | 0.067 | <0.01 |
| DSS accumulation | 0.0004 | m year$^{-1}$ | -0.0003 | 0.3 |





**Appendix C: Multiple linear regression models**

The summaries of the coefficients and model fit statistics for the linear models for 1979-2015 (Table C1, C2), 1957-2015 (Table C3, C4) and 1948-2015 (Table C5, C6) are shown here. The linear models use the annual frequency of selected synoptic types
to estimate DSS ice core annual accumulation. Note that the synoptic types used in each model were selected using stepwise model selection by AIC (stepAIC function in R).

**Table C1.** Coefficients of linear model for 1979-2015

Significance codes: 0 '***' 0.001 '**' 0.01 '*' 0.05 '.' 0.1 ' ' 1

|             | Estimate  | Std. Error | t value | Pr(>\|t\|)  |
| ----------- | --------- | ---------- | ------- | --------- |
| (Intercept) | 0.187052  | 0.235192   | 0.795   | 0.4327    |
| n4          | 0.003993  | 0.002091   | 1.909   | 0.0658 .  |
| n5          | 0.006343  | 0.002787   | 2.276   | 0.0301 *  |
| n8          | 0.004347  | 0.002215   | 1.963   | 0.0590 .  |
| n10         | 0.004626  | 0.002345   | 1.973   | 0.0578 .  |
| n12         | 0.003290  | 0.001461   | 2.253   | 0.0318 *  |
| n1          | -0.003597 | 0.001988   | -1.809  | 0.0804 .  |

**Table C2.** Model fit statistics for 1979-2015

|                         | Value  | DF     | p-value  |
| ----------------------- | ------ | ------ | -------- |
| Residual standard error | 0.1264 | 30     |          |
| Multiple R-squared      | 0.4218 |        |          |
| Adjusted R-squared      | 0.3062 |        |          |
| F-statistic             | 3.648  | 6, 30  | 0.007731 |

**Table C3.** Coefficients of linear model for 1957-2015

Significance codes: 0 '***' 0.001 '**' 0.01 '*' 0.05 '.' 0.1 ' ' 1

|             | Estimate  | Std. Error | t value | Pr(>\|t\|)       |
| ----------- | --------- | ---------- | ------- | -------------- |
| (Intercept) | 1.158062  | 0.093420   | 12.396  | $< 2e-16$ ***  |
| n1          | -0.005035 | 0.001441   | -3.493  | 0.000949 ***   |
| n9          | -0.002288 | 0.000691   | -3.311  | 0.001644 **    |
| n11         | -0.004477 | 0.001361   | -3.290  | 0.001753 **    |



**Table C4.** Model fit statistics for 1957-2015

|  | Value | DF | p-value |
|---|---|---|---|
| Residual standard error | 0.1244 | 55 | |
| Multiple R-squared | 0.2792 | | |
| Adjusted R-squared | 0.2398 | | |
| F-statistic | 7.1 | 3, 55 | 0.0004075 |

**Table C5.** Coefficients of linear model for 1948-2015

Significance codes: 0 '***' 0.001 '**' 0.01 '*' 0.05 '.' 0.1 ' ' 1

|  | Estimate | Std. Error | t value | Pr(>|t|) |
|---|---|---|---|---|
| (Intercept) | -26.95619 | 14.04867 | -1.919 | 0.0602 . |
| n1 | 0.07189 | 0.03851 | 1.867 | 0.0673 . |
| n2 | 0.08172 | 0.03925 | 2.082 | 0.0420 * |
| n3 | 0.07669 | 0.03857 | 1.988 | 0.0518 . |
| n4 | 0.07759 | 0.03859 | 2.011 | 0.0493 * |
| n5 | 0.07687 | 0.03847 | 1.998 | 0.0506 . |
| n6 | 0.07807 | 0.03838 | 2.034 | 0.0468 * |
| n7 | 0.07416 | 0.03799 | 1.952 | 0.0560 . |
| n8 | 0.07580 | 0.03842 | 1.973 | 0.0535 . |
| n9 | 0.07421 | 0.03842 | 1.932 | 0.0586 . |
| n10 | 0.07482 | 0.03836 | 1.950 | 0.0563 . |
| n11 | 0.07456 | 0.03853 | 1.935 | 0.0581 . |
| n12 | 0.07543 | 0.03826 | 1.972 | 0.0537 . |

**Table C6.** Model fit statistics for 1948-2015

|  | Value | DF | p-value |
|---|---|---|---|
| Residual standard error | 0.1264 | 55 | |
| Multiple R-squared | 0.3181 | | |
| Adjusted R-squared | 0.1694 | | |
| F-statistic | 2.139 | 12, 55 | 0.02878 |



*Author contributions.* This study was based on a University of Tasmania Bachelor of Marine and Antarctic Science Honours research project undertaken by MN and conceived by DU and TV. MN led the study and performed the data analyses. DU provided the code for the SOM algorithm, which was adapted by MN. Interpretation of results and writing the manuscript was led by MN with substantial contributions from DU and TV.


*Competing interests.* The authors claim no competing interests

*Acknowledgements.* *Financial Support* This study was supported by the Australian Government's Antarctic Science Collaboration Initiative (ASCI000002) through funding to the Australian Antarctic Program Partnership. This study contributes to an Australian Research Council (ARC) Discovery Project (DP220100606) to TRV. DGU was supported by the ARC Special Research Initiative Australian Centre of Ex-
cellence in Antarctic Science (SR200100008), ARC Discovery Project DP220100606 and ARC Centre of Excellence for Climate Extremes (CE170100023).



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
