# Peer review of "Evaluating the Twentieth Century Reanalysis Version 3 with synoptic typing and an East Antarctic ice core accumulation record"

_Climate of the Past, 2024_

## Referee Comment (RC3)

Comments on Nilssen et al. "**Evaluating the Twentieth Century Reanalysis Version 3 with synoptic typing and East Antarctic ice core accumulation**" submitted to *Climate of the Past*

Owing to limited weather records before the satellite era, understanding long-term variability and inter-decadal pattern in synoptic systems over East Antarctica is challenging. This study evaluated the ability of the Twentieth Century Reanalysis project to reproduce the synoptic conditions associated with increased precipitation at Law Dome since 1948, using daily 500 hPa geopotential height anomalies and the annual snowfall accumulation record from the ice core. The results indicate that this Reanalysis can reliably represent the meridional weather conditions of increased precipitation at Law Dome before the satellite era, and thus extends the time span of available materials for analyzing weather conditions for this region.

I appreciate the objective of this paper, and I am interested in the results and conclusions. However, there are still several issues to be clarified in this study. I recommend that this manuscript needs a major revision before published.

Major points:

1. Since the authors used Twentieth Century Reanalysis Version 3 to perform this study, I have a major concern on the reliability of the data. Especially, the data series for this atmospheric reanalysis may have suffered a "jump" at the ice sheet scale at the beginning of the satellite era. The authors should add some works to prove that it is reliable at least on regional scales (or at Law Dome). This is very important for the analyze, as the major results and conclusions are relied on the 20CRv3 data.

2. I suggest that the second part of the manuscript should be changed to "Data and Methods", and that its content needs to undergo a substantial re-organization to make it more coherent. For instance, the headings of 2.1 and 2.3 stand for "Data" rather than "Methods", and 2.4 includes too much information that is not relevant to the heading, such as the division of the period, and the title of 2.3 does not emphasize the classification of precipitation events. The authors should try to separate the description of the data and methods, and introduce each section specifically, such as "snowfall accumulation record from the Law Dome DSS; Twentieth Century Reanalysis version 3…".

3. The authors claim that the ice core record shows high accumulation rates and seasonality at Law Dome (L76-78), so I am concerned about the possibility of extracting seasonal climate signals (synoptic systems and accumulation) from the ice core record. This would not only enable assessing the reliability of the 20CR on a timescale with higher frequency, but also contribute to understanding the seasonal variability of synoptic patterns affecting the Law Dome.

4. Surface ablation rarely occurs over most of the Antarctic ice sheet, so snowfall accumulation is contributed mainly by precipitation. However, Law Dome is located in the Antarctic coastal region. Studies have been done to show that these areas near the coast are threatened by rainfall from extreme events such as atmospheric rivers. I would therefore suggest that you should distinguish the precipitation pattern (rainfall or snowfall, they have almost opposite effects on accumulation) in this study, rather than comparing precipitation directly to the accumulation from ice core record. Or, another approach is to confirm that rainfall-derived melting or snow blowing is not sufficient to have a significant effect on the inter-annual variability of snow accumulation at Law Dome.

5. Section 3.6 "Linear model estimates of ice core annual accumulation

from synoptic typing": The description in this section is too short and the authors should have described it in more detail.

6. Although this paper investigated synoptic types on a regional scale, the study relied on ice core records from the Law Dome, so it is inappropriate to show "East Antarctic ice core accumulation" in the title, and I suggest changing it to the "Law Dome". East Antarctica covers a much larger spatial area not studied by this paper, and a single ice core record may not be strongly spatially representative. The authors also mention in the description of L277-279 that the accumulation record will not appear in the Law Dome DSS when the location of the blocking is slightly offset. Therefore, much of this study is not actually representative of East Antarctica.

7. There are some technical corrections in the manuscript, such as the lack of a uniform format for the minus sign "-". In section 3.1, authors sometimes label p<..., sometimes labeled p=.... The authors need to re-check and re-edit them.

Minor points:

1. Please check the units of potential height in the Figures.

2. L35: please add the references, such as Zhang et al., 2018; Wang et al., 2020 (which has been presented in the references

3. L112: "The 90th and 99th percentile of 20CRv3 daily precipitation at Law Dome was calculated". How was the base period for defining extreme events chosen? Extreme precipitation calculated based on percentile thresholds will be very dependent on the selection of base period.

4. Figure 2 and Section 3.1: Please plot the linear trend of the two data series in Figure 2, respectively.

5. L177-179: What are the quantitative standards for dividing the weather types? Type 2 also seems to dominate by meridional, despite the blocking

high not landing on the ice sheet.

6. The discussion of the relationship between annual frequency of synoptic types and DSS accumulation is relevant and needs to be reflected in the Conclusions and Abstract.

References:

Zhang Y, Wang Y, Huai B, Ding M, Sun W. Skill of the two 20th century reanalyses in representing Antarctic near-surface air temperature. Int J Climatol. 2018; 38:4225–4238. https://doi.org/10.1002/joc.5563

---

## Author Comment (AC1)

Author responses to referee comment 1 - Jesper Sjolte

In this study Nilsson et al. investigate the cohesion between the weather variability and precipitation rates in the 20th Century Reanalysis v3 (20CRv3) compared to the accumulation at the Dome Summit South (DSS) drill site, Law Dome. The sparse observational data makes it challenging to constrain reanalysis products for this region and the authors use self-organizing maps to produce weather patterns and find the patterns which are correlated with reanalysis precipitation at DSS. Nilsson et al. then compares the weather patterns and reanalysis precipitation with the annual accumulation rates from the DSS ice core. The main conclusion is the 20CRv3 performs very well after 1948 when an adequate amount of data is entering the reanalysis. The authors go on to suggest guidelines for comparing reanalysis products with Antarctic ice core data and how to improve the reanalysis data in regions with sparse observations.

General comments.

Overall, I find this study to be interesting and relevant, and the results and conclusions contribute to an area which is sort of a white spot on the map. The science question is clear and analysis, as well as, figures, data and method description are generally well executed.

There are, however, a bit of work to be done before the manuscript is fully publishable. I have quite a few comments regarding clarity and readability, and not the least structure of the manuscript. I find that there is a lot of mixing of the results and discussion section, and basically the results section should be longer and the discussion shorter. All of the description and explanation connected to Figure 4, 5 and 6, as wells as, Table 3 should be in the results section. See also detailed comments below regarding this. Furthermore, I think writing in present tense when describing analysis done in the paper is more appropriate. Historical facts and events can be referred to in past tense.

Thank you for these comments. We will move the suggested sections (descriptions and explanations of Figs 4, 5, 6, Table 3, as well as lines from the detailed comments below) from the discussion into the results. We will move some of the description of the weather systems/synoptic patterns that are associated with high/extreme and low/zero precipitation from section 4.2 to section 3.5. We note that it is sometimes a little difficult to separate results and discussion in a descriptive context (such as the self-organising map descriptions) however, we will re-organise the above sections and then review for clarity and readability.

We will also rephrase the descriptions of the analysis to be in present tense.

I do wonder if a seasonal signal can be extracted from the ice core given the high accumulation rate, and the availability of high-resolution isotope and impurity data. This could give further insights to the seasonal variability of weather patterns in the region and also regarding seasonal trends in accumulation. Maybe a point for the discussion?

Thank you for this comment. This is an important point, and it is one of the main motivations behind this study. At this point we believe this work is the initial step along the way to being able to identify and then statistically verify 'within year' signals, hopefully at the seasonal scale, which would enable the demarcation of seasonal boundaries in the ice core data. This work is the first step, because we believe this

demarcation will need to be valid over a longer period of reliable data than just the satellite era (although the satellite era will obviously always remain the more reliable verification period). We will add details about the ongoing work we are conducting in this area to try to locate seasonal boundaries and / or synoptic scale proxy signals in high resolution East Antarctic ice cores, primarily a study on stratigraphic markers in the Law Dome ice core (Zhang et al., 2023) and the implications from this work.

Detailed comments.

L12 "with increased precipitation" and I suppose thus also decreased precipitation. Suggestion: "variability of precipitation amount"

This is a good point, since low/zero precipitation patterns are also important for precipitation variability. We will rephrase as "precipitation variability" instead of "increased precipitation".

L24 "These weather systems have changed in frequency over the satellite era ". Suggestion: "The occurrence of these weather systems have changed in frequency over the satellite era ".

We will reword based on your suggestion

L30 "satellite era, and include" suggestion: "satellite era. The most recent reanalysis products include".

We will reword based on your suggestion

L34-35 "These reanalyses are generally considered to perform poorly " a reference or two to support this would be in order.

We will add these references to support this:

Schneider, D. P. and Fogt, R. L.: Artifacts in Century-Length Atmospheric and Coupled Reanalyses Over Antarctica Due To Historical Data Availability, Geophysical Research Letters, 45, 964–973, https://doi.org/10.1002/2017GL076226, _eprint: https://onlinelibrary.wiley.com/doi/pdf/10.1002/2017GL076226, 2018.

Wang, Y., Hou, S., Ding, M., and Sun, W.: On the performance of twentieth century reanalysis products for Antarctic snow accumulation, Climate Dynamics, 54, 435–455, https://doi.org/10.1007/s00382-019-05008-4, 2020.

Zhang, Y., Wang, Y., Huai, B., Ding, M., Sun, W.: Skill of the two 20th century reanalyses in representing Antarctic near-surface air temperature, Int J Climatol, 38, 4225–4238, https://doi.org/10.1002/joc.5563, 2018.

L75 Add "Dome Summit South" before DSS.

We will correct this

L77 "0.69 metres ice equivalent from frequent cyclonic incursions " I suggest to make a full stop before *from* and explain about the processes forming the precipitation in the next sentence.

We propose to comprehensively re-write this section for clarity, including adding further information which will address the comments here and below relating to lines 77-80. This will include information on how the DSS snowfall accumulation rate record was developed, which includes aspects of snow densification, layer thinning and horizontal advection at the site (both over the satellite era, and also for the past 2000 years). We will also discuss in more detail how annual layers are identified at Law Dome during layer counting, volcanic alignment with known events, and current understanding of the mean seasonal variations in trace chemical records at Law Dome. We will include details of the numerous prior publications which have developed the Law Dome chronology, and the derived accumulation rate products for this site.

L77 "annual accumulation rate of 0.69 metres ice equivalent" I found no description how the accumulation is converted to ice equivalent. How is firnification and ice flow treated? Also, is evaporation and issue when comparing accumulation and reanalysis precipitation? I didn't found any mention of this. Maybe not an issue for Law Dome, but for other sites with low accumulation. Could be a point in the discussion.

We will rewrite this section to include more detail on how the snowfall accumulation record was developed (see comment above).

Regarding evaporation, we have no evidence from prior studies that evaporation / sublimation is a major or significant component of the variability in annual snowfall accumulation recorded at Law Dome. Law Dome is a high snowfall, wet deposition site (Roberts et al., 2015; Morgan et al., 1997) and the bulk of the snow that arrives at the site is buried in subsequent snowfall events, without relatively minor impacts from sublimation or wind erosion (e.g. McMorrow et al., 2001; 2004). We will note this.

L77-78 "produces seasonally varying annual layers" how can annual layers be seasonally varying? Please rephrase.

We will reword this section as per above.

L79 "Annual layers are identified" is this done in this study or in the studies you refer to?

We will reword this section as per above.

L80 "and validated against known volcanic eruptions" It's the time scale i.e., dating, which is validated using volcanic eruptions as tie points.

We will reword this section as per above.

L93 "applied to weather and climate applications" I think it should be "applied in", but maybe better to reformulate and avoid using both "applied" and "applications".

We will rephrase this as:

"and how it has been used for weather and climate applications, can be found in other studies"

L104-105 Maybe the correlations "can be expected" to be lower?

Agreed, we will rephrase

L107 "associated with increased precipitation at Law Dome " if you get which patters give increased precipitation, you also get the patterns that cause low precipitation. I think it is more intuitive to generally formulate that you want to establish the relation between the SOM patterns and precipitation amount at Law Dome.

We agree using the term 'variability' is far more descriptive here, and will rephrase as "precipitation variability" instead of "increased precipitation". We will also ensure this is corrected elsewhere it may occur in the text.

L112 Some kind of introducing part of the sentence should be added here so that one understands it's a new topic. Something like "We then calculated" "In a next step"…

We will add a few words to introduce the next topic.

Table 1: The standard would be to have two significant digits for correlation and explained variance. And keep it consistent between text and table. It says "r = 0.8" in the text L144.

We will change the r and R^2 values to two significant figures, and keep it consistent with the text.

L145 For trends upper and confidence bounds can be more instructive than a p-value.

We will add 95% confidence intervals for the trends instead of the p values.

Section 3.2: All but the last sentence of this section starts with "the" which makes it very repetitive. Consider rewriting with more flow by adding a few words where it fits, e.g. "Similar to" "In contrast with".

We will rephrase to reduce repetition.

Figure 2: Unit for accumulation in caption "miceequivalent". Mice equivalent? I think making the DSS accumulation as a stair-type plot would improve the readability of the graph.

Apologies, it should be "metres ice equivalent or m.i.e.", which we will define in the caption.

We will also change the DSS accumulation to a stair plot structure as suggested (see revised figure below).

[Figure]

Figure 3: You could add the DSS site in the figure so the reader has something to hang onto.

This is a good suggestion, we will add the DSS site into the figure

L192-193 "Synoptic types 3, 4, 8 and 12 were associated with high precipitation at Law Dome " this is a central result. Add some text to explain why these weather patterns result in high precipitation att DSS and others not. You explain this in Section 4.2 but I think this is part of the results.

We will move most of section 4.2 to section 3.5 in the results and ensure there is good flow between section 3.5 and th discussion.

L193: Fig. 5 is referred to before Fig 4. Check so that all figures are ordered in the same way they appear in the text.

We will check the order of the figures and ensure the document is compiling correctly.

L202: Discussion. I my mind the discussion cannot start here. You are not done describing your results. You are allowed to discuss things while describing your results in the extent they are need to explain things. For example, you can say that your results depend on the number of observations going into the reanalysis, given the topic of your study there is nothing controversial about that. In your discussion you write about uncertainties, relation to other studies, add minor results that might frame your main results and finally an outlook for future studies or recommendations.

Thank you for these helpful suggestions. We will carefully restructure the results and discussion, so that there are more explanations in the results as per the details we have provided in the comments above.

Figure 6: Use 2 significant digits for R^2 in figure.

We will correct this.

L211 "that have been assimilated into ISPDv4.7 increase from" something wrong in this sentence.

Agreed, we will rephrase this sentence as:

"In the study area, the number of assimilated observations in ISPDv4.7 increased from the late 1940s (Compo et al., 2019). Some of the key locations are described here. "

L216 Byrd station also established in 1957. Although not in East Antarctica this also helps constrain the large-scale circulation around Antarctica.

This is true, although we would argue that many stations were established in 1956-1958 across Antarctica. However, we take the reviewers point, and will edit that sentence to say:

"The International Geophysical Year (July 1957- December 1958) also saw a large increase in meteorological observations in East Antarctica, as well as other parts of the continent (Wexler, 1956)."

L224-241 Keep this in discussion.

We will keep this in the discussion

Section 4.2 is mainly results.

We will move most of section 4.2 into the results as noted in the comments above.

L271-274 Results.

We will move this part into the results

L274-281 This is mainly Discussion. You might note that blowing snow is less of an issue at high-accumulation sites.

We will add that loss due to winds is less of an issue at high accumulation sites, by rephrasing the sentence in lines 279-281:

"Wind erosion is less of an issue at Law Dome compared to other sites due to an absence of frequent high winds, but there is evidence of snowfall events missing from the net accumulation record (McMorrow et al., 2001; Zhang et al., 2023)."

L281-293 Results.

We will review this section to ensure any relevant results are moved to section 3.6

L293-301 Discussion.

We will keep this in the discussion

References:

Mcmorrow, A., Ommen, T. D. V., Morgan, V., and Curran, M. A. J.: Ultra-high-resolution seasonality of trace-ion species and oxygen isotope ratios in Antarctic firn over four annual cycles, Annals of Glaciology, 39, 34–40, https://doi.org/10.3189/172756404781814609, 2004.

McMorrow, A. J., Curran, M. A. J., Ommen, T. D. V., Morgan, V., Pook, M. J., and Allison, I.: Intercomparison of firn core and meteorological data, Antarctic Science, 13, 329–337, https://doi.org/10.1017/S0954102001000463, 2001.

Morgan, V. I., Wookey, C.W, Li, J., van Ommen, T.D., Skinner, W., and Fitzpatrick, M.F: Site information and initial results from deep ice drilling on Law Dome, Antarctica, J. Glaciol., 43, 3–10, doi:10.3189/S0022143000002768, 1997.

Roberts, J., Plummer, C., Vance, T., van Ommen, T., Moy, A., Poynter, S., Treverrow, A., Curran, M., and George, S.: A 2000-year annual record of snow accumulation rates for Law Dome, East Antarctica, Climate of the Past, 11, 697–707, https://doi.org/10.5194/cp-11-697-2015, publisher: Copernicus GmbH, 2015.

Zhang, L., Vance, T. R., Fraser, A. D., Jong, L. M., Thompson, S. S., Criscitiello, A. S., and Abram, N. J.: Identifying atmospheric processes favouring the formation of bubble-free layers in the Law Dome ice core, East Antarctica, The Cryosphere, 17, 5155–5173, https://doi.org/10.5194/tc-17-5155-2023, publisher: Copernicus GmbH, 2023.

---

## Author Comment (AC3)

Comments on Nilssen et al. "Evaluating the Twentieth Century
Reanalysis Version 3 with synoptic typing and East Antarctic ice core
accumulation" submitted to Climate of the Past
Owing to limited weather records before the satellite era, understanding
long-term variability and inter-decadal pattern in synoptic systems over
East Antarctica is challenging. This study evaluated the ability of the
Twentieth Century Reanalysis project to reproduce the synoptic conditions
associated with increased precipitation at Law Dome since 1948, using
daily 500 hPa geopotential height anomalies and the annual snowfall
accumulation record from the ice core. The results indicate that this
Reanalysis can reliably represent the meridional weather conditions of
increased precipitation at Law Dome before the satellite era, and thus
extends the time span of available materials for analyzing weather
conditions for this region.
I appreciate the objective of this paper, and I am interested in the results
and conclusions. However, there are still several issues to be clarified in
this study. I recommend that this manuscript needs a major revision before
published.

Major points:
1. Since the authors used Twentieth Century Reanalysis Version 3 to
perform this study, I have a major concern on the reliability of the data.
Especially, the data series for this atmospheric reanalysis may have
suffered a "jump" at the ice sheet scale at the beginning of the satellite era.
The authors should add some works to prove that it is reliable at least on
regional scales (or at Law Dome). This is very important for the analyze,
as the major results and conclusions are relied on the 20CRv3 data.

Thank you for this comment. The reliability of 20CRv3 is certainly a major issue that needs
to be addressed. None of the twentieth century reanalyses use satellite data, but CERA-20C
does show a jump in P-E over the Antarctic ice sheet at the beginning of the satellite era,
while 20CR shows a similar jump in 1950 (Wang et al., 2020). We state in lines 49-54 that
we chose 20CR over ERA-20C for this study, as 20CR has been determined to be less
vulnerable to inhomogeneities due to changes in observation density, as well as the different
assimilation schemes. However, 20CRv3 does still have a significant increase in the number
of assimilated observations over time, and therefore significant changes in error and
reliability over time. We agree that determining the reliability at specific regions or points is
very important, which is why we believe our study, using a well understood and accurately
dated ice core record as comparison to reanalysis, contributes to this question.

We will add this sentence in the discussion (at the end of the paragraph that ends at line
230):

"However, evaporation is considered to be a minor contributor to variability at the Law Dome
site, and so is unlikely to have a major effect on this study (Roberts et al., 2015)."

2. I suggest that the second part of the manuscript should be changed to
"Data and Methods", and that its content needs to undergo a substantial reorganization to
make it more coherent. For instance, the headings of 2.1

and 2.3 stand for "Data" rather than "Methods", and 2.4 includes too much information that is not relevant to the heading, such as the division of the period, and the title of 2.3 does not emphasize the classification of precipitation events. The authors should try to separate the description of the data and methods, and introduce each section specifically, such as "snowfall accumulation record from the Law Dome DSS; Twentieth Century Reanalysis version 3…".

Thank you for this comment. We will reorganise these sections to improve coherency. The headings will be as follows:

2 Data and Methods
2.1 Data Used
2.1.1 Twentieth Century Reanalysis version 3
2.1.2 Snowfall accumulation record from Law Dome DSS ice core
2.2 Methods
2.2.1 Self-organising map inputs and evaluation
2.2.2 20CRv3 daily precipitation and synoptic types
2.2.3 Classification of precipitation events
2.2.4 Division of time period and correlation between annual precipitation and accumulation
2.2.5 Regression analysis of snowfall accumulation using synoptic types

3. The authors claim that the ice core record shows high accumulation rates and seasonality at Law Dome (L76-78), so I am concerned about the possibility of extracting seasonal climate signals (synoptic systems and accumulation) from the ice core record. This would not only enable assessing the reliability of the 20CR on a timescale with higher frequency, but also contribute to understanding the seasonal variability of synoptic patterns affecting the Law Dome.

Indeed, understanding the seasonal signals and seasonal variability at Law Dome (and by extension, other high snowfall ice core sites) is really the driving purpose of this study. We think that being able to unlock some of the seasonal variability signals from ice core records like Law Dome, either in sea salt (e.g. Vance et al., 2013; Udy et al., 2024), stratigraphy (Zhang et al., 2023) or stable water isotopes (e.g. Jackson et al., 2023) would be a hugely powerful tool in understanding how synoptic scale processes have changed in the past. However, disentangling these signals at less than annual scales in ice core records is currently still challenging, and relies on very detailed and precise dating of the annual layers. This study is another step in that direction, because it gives us evidence that we can explore the interplay between synoptic types and changes in the ice core accumulation, chemistry and stratigraphy prior to the satellite era - e.g., we now have some confidence to use 20CR to explore the seasonal cycle of snowfall at Law Dome for over 60 years. This will help us understand over the longer term when snow falls and how episodic it is - a key piece of information in then exploring any chemical or stratigraphic markers associated with seasonal snowfall. We will add some sentences to the end of the introduction and in the discussion to ensure our purpose with this study is clear to the reader.

4. Surface ablation rarely occurs over most of the Antarctic ice sheet, so snowfall accumulation is contributed mainly by precipitation. However, Law Dome is located in the Antarctic coastal region. Studies have been done to show that these areas near the coast are threatened by rainfall from extreme events such as atmospheric rivers. I would therefore suggest that you should distinguish the precipitation pattern (rainfall or snowfall, they

have almost opposite effects on accumulation) in this study, rather than comparing precipitation directly to the accumulation from ice core record. Or, another approach is to confirm that rainfall-derived melting or snow blowing is not sufficient to have a significant effect on the inter-annual variability of snow accumulation at Law Dome.

We agree that there seems to be an increasing threat of rainfall in coastal regions, and these events are increasingly being catalogued not only on the Antarctic Peninsula, but also coastal East Antarctica. However, rainfall events at Law Dome are, at this stage, still vanishingly rare, and we know this for sure because rainfall on a snowpack leaves a very clear signal of frozen liquid water that has percolated into the snowpack, as well as a disrupted oxygen isotope record. Law Dome summit is certainly a coastal location, however its elevation (1,370 metres) means its precipitation type remains only snowfall (and a small fraction of diamond dust/clear sky precipitation), because mean annual temperatures are quite low (~-22 ℃). Ensuring non-liquid precipitation only was a key aspect of the original site selection for Law Dome (and this is usually the case for the site selection of other coastal records as well). Additionally, we have no evidence from our numerous overlapping surface and deep ice core records from Dome Summit South (e.g. compiled in Jong et al., 2022) or from stratigraphic studies at the DSS site, that rainfall events occur (Zhang et al., 2023), and we know of only one instance of a melt layer (from warmer than average temperatures and high solar radiation) being observed in an ice core drilled on the eastern flank of Law Dome at a lower elevation than DSS (Pers. Comm. David Etheridge 2023). Thus, we don't think there is much point in differentiating rainfall and snowfall at the DSS site, as no rainfall has historically occurred (and we know this as we would clearly see the resulting percolated melt layers in our ice core records).

We will add a sentence to the methods and DSS site description to note the above.

5. Section 3.6 "Linear model estimates of ice core annual accumulation from synoptic typing": The description in this section is too short and the authors should have described it in more detail.

We will move some of the detail about the regression models from the discussion into section 3.6.

6. Although this paper investigated synoptic types on a regional scale, the study relied on ice core records from the Law Dome, so it is inappropriate to show "East Antarctic ice core accumulation" in the title, and I suggest changing it to the "Law Dome". East Antarctica covers a much larger spatial area not studied by this paper, and a single ice core record may not be strongly spatially representative. The authors also mention in the description of L277-279 that the accumulation record will not appear in the Law Dome DSS when the location of the blocking is slightly offset. Therefore, much of this study is not actually representative of East Antarctica.

While it is true that we use only one ice core record to 'groundtruth' our findings, we disagree that our title is inappropriate, since we are using the reanalysis to evaluate whether we can look at regional synoptic scale variability with relevance to most of East Antarctica. The synoptic variability, and the ability for 20CR to discern precipitation types (e.g. high and extreme precipitation) is the important finding, as this will allow us and others to have confidence in using a reanalysis that is longer than the satellite era for the East Antarctic region. The use of Law Dome is more to check whether the ice core accumulation record is

also representative. However, we think the results from our study are more likely to be utilised for the longer, regional synoptic typing dataset, and this implies regional rather than local applicability. We propose to change the title slightly to:

"Evaluating the Twentieth Century Reanalysis Version 3 with synoptic typing and an East Antarctic ice core accumulation record."

7. There are some technical corrections in the manuscript, such as the lack of a uniform format for the minus sign "-". In section 3.1, authors sometimes label p<..., sometimes labeled p=.... The authors need to recheck and re-edit them.

We will check these and ensure they are consistent. For the trends in section 3.1 and 3.5, we will replace the p values with 95% confidence intervals.

Minor points:
1. Please check the units of potential height in the Figures.

We have checked this, and these are correct as submitted. Figure 1 shows the mean 500 hPa geopotential height in metres, while Figure 3 shows the 500 hPa geopotential height anomaly in metres. The actual height vs the anomaly may have caused some confusion.

2. L35: please add the references, such as Zhang et al., 2018; Wang et al., 2020 (which has been presented in the references

We will add the following references:

Schneider, D. P. and Fogt, R. L.: Artifacts in Century-Length Atmospheric and Coupled Reanalyses Over Antarctica Due To Historical Data Availability, Geophysical Research Letters, 45, 964–973, https://doi.org/10.1002/2017GL076226, _eprint: https://onlinelibrary.wiley.com/doi/pdf/10.1002/2017GL076226, 2018.

Wang, Y., Hou, S., Ding, M., and Sun, W.: On the performance of twentieth century reanalysis products for Antarctic snow accumulation, Climate Dynamics, 54, 435–455, https://doi.org/10.1007/s00382-019-05008-4, 2020.

Zhang, Y., Wang, Y., Huai, B., Ding, M., Sun, W.: Skill of the two 20th century reanalyses in representing Antarctic near-surface air temperature, Int J Climatol, 38, 4225–4238, https://doi.org/10.1002/joc.5563, 2018.

3. L112: "The 90th and 99th percentile of 20CRv3 daily precipitation at Law Dome was calculated". How was the base period for defining extreme events chosen? Extreme precipitation calculated based on percentile thresholds will be very dependent on the selection of base period.

Thank you for picking this up. The base period was chosen to be 1900-2015, which we will add to the text. The 90th and 99th percentiles of daily precipitation would have been higher if we had chosen a different base period (eg 1950-2015 or 1979-2015), which would have reduced the amount of annual 20CRv3 precipitation from high and extreme days, and reduced the total number of high and extreme days. Either way, we would still see the

increase of high and extreme precipitation days from around 1950. We will add some explanation in the methods about this, and when we discuss the corresponding results.

4. Figure 2 and Section 3.1: Please plot the linear trend of the two data series in Figure 2, respectively.

We would prefer not to do this, because we think that adding trend lines to Figure 2 would make the plot busy, and importantly, would be somewhat misleading and cause some confusion. Figure 2 is to demonstrate the agreement (or lack thereof, in the first half of the 20th century) between the total annual 20CRv3 precipitation at Law Dome, and the annual snowfall accumulation at Law Dome, along with the change in 20CRv3 precipitation attributed to high/extreme precipitation days at specific points through the 20th century. The overall trends in the two data series are discussed and shown in Table 1, but the point of this figure is that linear trends would be an insuffiecient way to examine or make inferences about the variability through time, as the figure shows not only step changes at different points (e.g. 1948 and 1958) but also trend changes (e.g. 20th century Law Dome precipitation compared to satellite era Law Dome accumulation) . These changes are discussed more comprehensively in sections 3.2, 3.5 and 4.1.

5. L177-179: What are the quantitative standards for dividing the weather types? Type 2 also seems to dominate by meridional, despite the blocking high not landing on the ice sheet.

Segregating or dividing the weather types by dominant atmospheric pattern is a subjective process when using this type of analysis, in this and other studies. This means that different interpretations are possible. However, while type 2 does appear more meridional than the other mixed types, we classified it as mixed because the geopotential height anomalies are much weaker than those that are observable in the types we classified as meridional (1, 3, and 4). For this study, we are interested in meridional transport of moisture to Law Dome, and by extension other regions of East Antarctica from other synoptic types. The kinds of weather seen on type 2 days would not be strongly meridional from the perspective of additional moisture transport to the region including Law Dome. We will make this distinction in the text (section 3.4). As the reviewer mentions, the block does not extend to the ice sheet, and we know this is critical in changing the moisture transport regime to the ice sheet (e.g. Jackson et al., 2023, Pohl et al., 2021), so we think that type 2 should remain a mixed rather than meridional type.

We will add "favourable for meridional transport of moisture to East Antarctica" to line 177.

6. The discussion of the relationship between annual frequency of synoptic types and DSS accumulation is relevant and needs to be reflected in the Conclusions and Abstract.

We will further discussion of the relationship between the annual frequency of synoptic types and DSS accumulation to the conclusions and abstract.

References:
Zhang Y, Wang Y, Huai B, Ding M, Sun W. Skill of the two 20th century reanalyses in representing Antarctic near-surface air temperature. Int J Climatol. 2018; 38:4225–4238. https://doi.org/10.1002/joc.5563

References:

Jackson, S. L., Vance, T. R., Crockart, C., Moy, A., Plummer, C., and Abram, N. J.: Climatology of the Mount Brown South ice core site in East Antarctica: implications for the interpretation of a water isotope record, Climate of the Past, 19, 1653–1675, https://doi.org/10.5194/cp-19-1653-2023, publisher: Copernicus GmbH, 2023.

Jong, L. M., Plummer, C. T., Roberts, J. L., Moy, A. D., Curran, M. A. J., Vance, T. R., Pedro, J. B., Long, C. A., Nation, M., Mayewski, P. A., and van Ommen, T. D.: 2000 years of annual ice core data from Law Dome, East Antarctica, Earth System Science Data, 14, 3313–3328, https://doi.org/10.5194/essd-14-3313-2022, publisher: Copernicus GmbH, 2022.

Pohl, B., Favier, V., Wille, J., Udy, D. G., Vance, T. R., Pergaud, J., Dutrievoz, N., Blanchet, J., Kittel, C., Amory, C., Krinner, G., and Codron, F.: Relationship Between Weather Regimes and Atmospheric Rivers in East Antarctica, Journal of Geophysical Research: Atmospheres, 126, e2021JD035 294, https://doi.org/10.1029/2021JD035294, _eprint: https://onlinelibrary.wiley.com/doi/pdf/10.1029/2021JD035294, 2021.

Roberts, J., Plummer, C., Vance, T., van Ommen, T., Moy, A., Poynter, S., Treverrow, A., Curran, M., and George, S.: A 2000-year annual record of snow accumulation rates for Law Dome, East Antarctica, Climate of the Past, 11, 697–707, https://doi.org/10.5194/cp-11-697-2015, publisher: Copernicus GmbH, 2015.

Udy, D.G., Vance, T.R., Kiem, A.S., Holbrook, N.J, and Abram, N.: Australia's 2019/20 Black Summer fire weather exceptionally rare over the last 2000 years, Commun Earth Environ, 5, 317, https://doi.org/10.1038/s43247-024-01470-z, 2024.

Vance, T. R., Ommen, T. D. v., Curran, M. A. J., Plummer, C. T., and Moy, A. D.: A Millennial Proxy Record of ENSO and Eastern Australian Rainfall from the Law Dome Ice Core, East Antarctica, Journal of Climate, 26, 710–725, https://doi.org/10.1175/JCLI-D-12-00003.1, publisher: American Meteorological Society Section: Journal of Climate, 2013.

Wang, Y., Hou, S., Ding, M., and Sun, W.: On the performance of twentieth century reanalysis products for Antarctic snow accumulation, Climate Dynamics, 54, 435–455, https://doi.org/10.1007/s00382-019-05008-4, 2020.

Zhang, L., Vance, T. R., Fraser, A. D., Jong, L. M., Thompson, S. S., Criscitiello, A. S., and Abram, N. J.: Identifying atmospheric processes favouring the formation of bubble-free layers in the Law Dome ice core, East Antarctica, The Cryosphere, 17, 5155–5173, https://doi.org/10.5194/tc-17-5155-2023, publisher: Copernicus GmbH, 2023.

---

## Author Response (AR1)

**Author responses to reviewers**

We thank both reviewers for their very helpful and constructive comments, which have greatly improved our manuscript.

In addition to the changes noted below, we have also made minor textual changes to improve readability and to fix typographical errors. We have also revised the colour scheme in Figure 4(c).

In our responses, all page and section numbers refer to updated manuscript, unless otherwise specified. Reviewer comments are in black, author responses are in blue.

**Author responses to referee comment 1 - Jesper Sjolte**

In this study Nilsson et al. investigate the cohesion between the weather variability and precipitation rates in the 20th Century Reanalysis v3 (20CRv3) compared to the accumulation at the Dome Summit South (DSS) drill site, Law Dome. The sparse observational data makes it challenging to constrain reanalysis products for this region and the authors use self-organizing maps to produce weather patterns and find the patterns which are correlated with reanalysis precipitation at DSS. Nilsson et al. then compares the weather patterns and reanalysis precipitation with the annual accumulation rates from the DSS ice core. The main conclusion is the 20CRv3 performs very well after 1948 when an adequate amount of data is entering the reanalysis. The authors go on to suggest guidelines for comparing reanalysis products with Antarctic ice core data and how to improve the reanalysis data in regions with sparse observations.

**General comments.**

Overall, I find this study to be interesting and relevant, and the results and conclusions contribute to an area which is sort of a white spot on the map. The science question is clear and analysis, as well as, figures, data and method description are generally well executed.

There are, however, a bit of work to be done before the manuscript is fully publishable. I have quite a few comments regarding clarity and readability, and not the least structure of the manuscript. I find that there is a lot of mixing of the results and discussion section, and basically the results section should be longer and the discussion shorter. All of the description and explanation connected to Figure 4, 5 and 6, as wells as, Table 3 should be in the results section. See also detailed comments below regarding this. Furthermore, I think writing in present tense when describing analysis done in the paper is more appropriate. Historical facts and events can be referred to in past tense.

Thank you for these comments. We have moved the descriptions and explanations of Figs 4, 5, 6, Table 3, as well as lines from the detailed comments below from the discussion into the results, where appropriate. We have moved some of the description of the weather systems/synoptic patterns that are associated with high/extreme and low/zero precipitation from section 4.2 to section 3.5. We note that it is sometimes a little difficult to separate results and discussion in a descriptive context (such as the self-organising map descriptions) however, we have reorganised the above sections for clarity and readability. See comments below for specific sections and line numbers.

We have also rephrased the descriptions of the analysis to be in present tense where this makes grammatical sense, e.g where we are discussing the results and conclusions of our study and their implications.

I do wonder if a seasonal signal can be extracted from the ice core given the high accumulation rate, and the availability of high-resolution isotope and impurity data. This could give further insights to the seasonal variability of weather patterns in the region and also regarding seasonal trends in accumulation. Maybe a point for the discussion?

Thank you for this comment. This is an important point, and it is one of the main motivations behind this study. At this point we believe this work is the initial step along the way to being able to identify and then statistically verify 'high frequency' or even synoptic scale proxy signals, which would enable the demarcation of seasonal boundaries in the ice core data. This work is the first step, because we believe this demarcation will need to be valid over a longer period of reliable data than just the satellite era (although the satellite era will obviously always remain the more reliable verification period). We have added details about the ongoing work we are conducting in this area to try to locate seasonal boundaries and / or synoptic scale proxy signals in high resolution East Antarctic ice cores, primarily a study on stratigraphic markers in the Law Dome ice core (Zhang et al., 2023) and the implications from this work. Lines 352-354.

"In addition, this study will greatly contribute to our ability to develop and understand proxies of seasonal and episodic accumulation in ice core records (e.g. see Zhang et al. (2023))."

Detailed comments.

L12 "with increased precipitation" and I suppose thus also decreased precipitation. Suggestion: "variability of precipitation amount"

This is a good point, since low/zero precipitation patterns are also important for precipitation variability. We have replaced "increased precipitation" with "precipitation variability" in line **12**, and wherever else it appeared in the text.

L24 "These weather systems have changed in frequency over the satellite era ". Suggestion: "The occurrence of these weather systems have changed in frequency over the satellite era ".

Reworded as suggested. Line 27.

L30 "satellite era, and include" suggestion: "satellite era. The most recent reanalysis products include".

Reworded as suggested. Line 34.

L34-35 "These reanalyses are generally considered to perform poorly " a reference or two to support this would be in order.

These references added to support this (line 40):

Schneider, D. P. and Fogt, R. L.: Artifacts in Century-Length Atmospheric and Coupled Reanalyses Over Antarctica Due To Historical Data Availability, Geophysical Research Letters, 45, 964–973, https://doi.org/10.1002/2017GL076226, \_eprint: https://onlinelibrary.wiley.com/doi/pdf/10.1002/2017GL076226, 2018. Wang, Y., Hou, S., Ding, M., and Sun, W.: On the performance of twentieth century reanalysis products for Antarctic snow accumulation, Climate Dynamics, 54, 435–455, https://doi.org/10.1007/s00382-019-05008-4, 2020.

Zhang, Y., Wang, Y., Huai, B., Ding, M., Sun, W.: Skill of the two 20th century reanalyses in representing Antarctic near-surface air temperature, Int J Climatol, 38, 4225–4238, https://doi.org/10.1002/joc.5563, 2018.

L75 Add "Dome Summit South" before DSS.

Now defined earlier in the text (line 78).

L77 "0.69 metres ice equivalent from frequent cyclonic incursions " I suggest to make a full stop before *from* and explain about the processes forming the precipitation in the next sentence.

We have comprehensively re-written this section (section 2.1.2, lines 84-114, pasted below) for clarity, including adding further information which addresses the comments here and below relating to lines 77-80 from the original manuscript. This includes information on how the DSS snowfall accumulation rate record has been developed, which includes aspects of firn densification and layer thinning at the site (both over the satellite era, and also for the past 2000 years). We also discuss in more detail how annual layers are identified at Law Dome during layer counting, volcanic alignment with known events, and the mean seasonal variations in trace chemical records at Law Dome. We include details of the numerous prior publications which have developed the Law Dome chronology, a brief assessment of volcanic date cross-matching with known eruptions, and the derived accumulation rate products for this site.

**"2.1.2 Annual snowfall accumulation record from the Law Dome (DSS) ice core**

We used the annual snowfall accumulation record from the Law Dome DSS ice core site in this study, which has been progressively developed over the past two decades (e.g. see Jong et al. (2022); Roberts et al. (2015); van Ommen and Morgan (2010) and references therein). The DSS site is situated in a coastal and predominantly wet deposition zone, and the ice core record has been regularly updated with short surface cores since the original deep drilling campaign was completed in the 1990s. Given its relatively high elevation (1,370 metres above sea level), DSS receives precipitation predominantly in the form of snowfall with an insignificant contribution from clear sky precipitation (diamond dust). We are not aware of any evidence of significant melt or rainfall events at DSS, despite rainfall being possible during extreme warm events along the East Antarctic coastline (Wille et al., 2024b, a). Any rainfall or melt events would appear as clear stratigraphic boundaries in the ice core record, and these have not been observed (van Ommen and Morgan, 1997; Zhang et al., 2023).

DSS receives relatively high annual snowfall compared to much of coastal East Antarctic, due to the interaction between frequent cyclonic incursions and the topography of the Dome (Mcmorrow et al., 2004). This high annual snowfall means seasonally varying trace impurity concentrations (e.g. sea salts, sulfate) and water stable isotope ratios ( $\overline{0}180$  and  $\overline{0}D$ ) can be relatively easily discerned. In combination, these seasonal variations allow the identification of annual horizons with a nominal date of January 10, allowing dating via annual layer counting (van Ommen and Morgan, 1997; McMorrow et al., 2001; Mcmorrow et al., 2004; Plummer et al., 2012; Jong et al., 2022). Additional dating accuracy is achieved via the identification of volcanic sulfate peaks, which are then cross-referenced to the dates of known global eruptions and compared to other East Antarctic and global ice core volcanic records (Plummer et al., 2012). Key eruptions observed in the DSS record over the time period of interest to this study include

Pinatubo (1991), Agung (1965), Krakotoa (1885) and Tambora (1816). The DSS layer counted record exhibits no dating error with the sulfate signatures of these well-documented eruptions (see table 3 in (Vance et al., 2024)).

The DSS annually resolved dataset (including annual accumulation) spans -11 to 2017 CE, which corresponds to around 800 metres deep to the surface at time of drilling in austral summer 2018 (see Jong et al. (2022)for detail and data access). The full DSS ice core reaches a depth of approximately 1,200 metres to bedrock (van Ommen and Morgan, 1997). Understanding of DSS site firn compaction and layer thinning has been progressively refined over the past two decades using firn core density estimates (calculated from core volume / weight measurements). Assuming steady state for depth profiles of both density (Sorge's law) and the vertical strain rate, Roberts et al. (2015) converted year boundary depths to ice equivalent depths to account for firn compaction, and then used a power-law vertical strain rate method to estimate annual snowfall accumulation rates. See Roberts et al. (2015) for a detailed explanation of the development of the DSS annual snowfall accumulation record and Jong et al. (2022) for the updated record used here. This study focuses on the most recent 120 years of this record. The mean annual snowfall accumulation rate derived using the above information at DSS is 0.69 metres ice equivalent (Crockart et al., 2021; Roberts et al., 2015)."

L77 "annual accumulation rate of 0.69 metres ice equivalent " I found no description how the accumulation is converted to ice equivalent. How is firnification and ice flow treated? Also, is evaporation and issue when comparing accumulation and reanalysis precipitation? I didn't found any mention of this. Maybe not an issue for Law Dome, but for other sites with low accumulation. Could be a point in the discussion.

We have rewritten this section to include more detail on how the snowfall accumulation record was developed (see comment above), lines 94-112. We have also greatly expanded section 2.1.2, which describes the development of the 2000 year annual accumulation record at Law Dome.

Regarding evaporation, we have no evidence from prior studies that evaporation / sublimation is a major or significant component of the variability in annual snowfall accumulation recorded at Law Dome. Law Dome is a high snowfall, wet deposition site (line 86) (Roberts et al., 2015; Morgan et al., 1997) and the bulk of the snow that arrives at the site is buried in subsequent snowfall events, with relatively minor impacts from sublimation or wind erosion (e.g. McMorrow et al., 2001; 2004).

L77-78 "produces seasonally varying annual layers " how can annual layers be seasonally varying? Please rephrase.

Reworded (see comment above), lines 94-97.

L79 "Annual layers are identified " is this done in this study or in the studies you refer to?

This is done in the studies we refer to. Reworded to make this clear (see comment above), lines 94-97.

L80 "and validated against known volcanic eruptions " It's the time scale i.e., dating, which is validated using volcanic eruptions as tie points.

Yes, this was poorly worded. Reworded with more detail, lines 98-103.

L93 "applied to weather and climate applications " I think it should be "applied in", but maybe better to reformulate and avoid using both "applied" and "applications".

Rephrased (lines 131-132): "and how it has been used for weather and climate applications, can be found in other studies"

L104-105 Maybe the correlations "can be expected" to be lower?

Agreed, we have rephrased as per your suggestion, line 143.

L107 "associated with increased precipitation at Law Dome " if you get which patters give increased precipitation, you also get the patterns that cause low precipitation. I think it is more intuitive to generally formulate that you want to establish the relation between the SOM patterns and precipitation amount at Law Dome.

We agree using the term 'variability' is far more descriptive here, and have rephrased as "precipitation variability" instead of "increased precipitation". We have also corrected it elsewhere in the text.

L112 Some kind of introducing part of the sentence should be added here so that one understands it's a new topic. Something like "We then calculated" "In a next step"...

We added "To differentiate between amounts of daily precipitation, we then calculated...", line 152.

Table 1: The standard would be to have two significant digits for correlation and explained variance. And keep it consistent between text and table. It says "r = 0.8" in the text L144.

We have changed the r and R^2 values to two significant figures, and kept it consistent with the text, line 186, table 1.

L145 For trends upper and confidence bounds can be more instructive than a p-value.

We replaced p-values with 95% confidence intervals for the trends, lines 187-190.

Section 3.2: All but the last sentence of this section starts with "the" which makes it very repetitive. Consider rewriting with more flow by adding a few words where it fits, e.g. "Similar to" "In contrast with".

We have rephrased to reduce repetition, and to add more detail. Lines 192-199.

"The 90th percentile of 20CRv3 daily precipitation at Law Dome is 5.05 mm day-1, and the 99th percentile is 13.4 mm day-1, based on the 1900-2015 period for consistency across the study period. The 90th percentile threshold from 20CRv3 over 1900-2015 is similar to the 90th percentile threshold calculated over 1979-2016 using RACMO2 (4.23 mm day-1) (Turner et al., 2019). Total annual precipitation from high precipitation days (90-99th percentile) increases from the late 1940s, with a further increase from the mid-late 1950s (Fig. 2). Precipitation that can be attributed to extreme precipitation days (over 99th percentile) increases from the mid-late 1950s, and a further increase from around 1980. Between 1900 and 1956, 9.4% of total precipitation came from high precipitation days, and 0.4% from extreme, and for 1957-2015, 45.7% of total precipitation came from high precipitation days, and 12.9% from extreme."

Figure 2: Unit for accumulation in caption "miceequivalent". Mice equivalent? I think making the DSS accumulation as a stair-type plot would improve the readability of the graph.

Apologies, it should be "metres ice equivalent", which we have defined in the caption.

We also updated the figure to include a stair plot for DSS accumulation as suggested (see revised Figure 2 below).

Figure 3: You could add the DSS site in the figure so the reader has something to hang onto.

**This is a good suggestion, we have added the DSS site into Figure 3.**

L192-193 "Synoptic types 3, 4, 8 and 12 were associated with high precipitation at Law Dome " this is a central result. Add some text to explain why these weather patterns result in high precipitation att DSS and others not. You explain this in Section 4.2 but I think this is part of the results.

We agree that this is a central result. We have restructured sections 3.5 and 4.2 to ensure adequate attention to this result and good flow between results and discussion. Lines 238-248.

"Synoptic types 3, 4, 8 and 12 are associated with above average precipitation at Law Dome (Fig. 5a, b). Synoptic types 3, 4 and 8 display strong positive geopotential height anomalies in the Tasman Sea region (Fig. 3), which represent anticyclonic blocking patterns that increase precipitation in East Antarctica (Scarchilli et al., 2011; Servettaz et al., 2020; Udy et al., 2021, 2022), and are often associated with atmospheric rivers (Pohl et al., 2021; Wille et al., 2021). Synoptic type 12, which is associated with high and extreme precipitation days (Fig. 5b), has a zonal negative height anomaly in the mid-latitudes, consistent with a negative SAM (SAM-) pattern. Previous studies have found that SAM- is associated with increased precipitation in the Law Dome region (Marshall et al., 2017). Type 12 in this study is comparable to SOM3 in Udy et al. (2021), which showed enhanced strength of polar easterlies and positive precipitation anomaly in the Law Dome region (Udy et al., 2022). This suggests that the precipitation associated with type 12 is predominantly orographic in nature, from moist air uplifted across Law Dome (Udy et al., 2021, 2022)."

L193: Fig. 5 is referred to before Fig 4. Check so that all figures are ordered in the same way they appear in the text.

We have checked the order of the figures and ensured the document is compiling correctly.

L202: Discussion. I my mind the discussion cannot start here. You are not done describing your results. You are allowed to discuss things while describing your results in the extent they are need to explain things. For example, you can say that your results depend on the number of observations going into the reanalysis, given the topic of your study there is nothing controversial about that. In your discussion you write about uncertainties, relation to other studies, add minor results that might frame your main results and finally an outlook for future studies or recommendations.

Thank you for these helpful suggestions. We have carefully restructured the results and discussion, so that there are more explanations in the results as per the details we have provided in the comments above and below.

Figure 6: Use 2 significant digits for R2 in figure.

We have corrected this.

L211 "that have been assimilated into ISPDv4.7 increase from" something wrong in this sentence.

Agreed, we have rephrased this sentence (lines 293-295)

"In the study area, the number of assimilated observations in ISPDv4.7 increased from the late 1940s (Compo et al., 2019). Some of the key locations that led to this increase in assimilated observations are described here."

L216 Byrd station also established in 1957. Although not in East Antarctica this also helps constrain the large-scale circulation around Antarctica.

This is true, although we would argue that many stations were established in 1956-1958 across Antarctica. However, we take the reviewers point, and have edited that sentence (lines 299-300)

"The International Geophysical Year (July 1957- December 1958) also saw a large increase in meteorological observations in East Antarctica, as well as other parts of the continent (Wexler, 1956)."

L224-241 Keep this in discussion.

We have kept this in the discussion.

Section 4.2 is mainly results.

We have rearranged and restructured section 4.2, see comment above. Lines 239-248 from the original section 4.2 are now in the results in section 3.5. Lines 327-337 now make up section 4.2, which now focuses on types 9 and 12, and SAM.

L271-274 Results.

We have moved this to the results, lines 256-259.

"Despite synoptic type 3 having the highest median daily precipitation (2.96 mm day-1 compared to the overall median of 1.79 mm day-1), the annual frequency of synoptic type 3 is not significantly correlated with the DSS ice core annual accumulation in any of the time periods tested. In contrast, synoptic type 3 is significantly correlated with 20CRv3 annual precipitation for only the 1900-2015 period."

L274-281 This is mainly Discussion. You might note that blowing snow is less of an issue at high-accumulation sites.

We have added that loss due to winds is less of an issue at high accumulation sites, by rephrasing the sentence in lines 345-347:

"Wind erosion is less of an issue at Law Dome compared to other sites due to an absence of frequent high winds, but there is evidence of snowfall events missing from the net accumulation record (McMorrow et al., 2001; Zhang et al., 2023)."

**L281-293 Results.**

We have reviewed this section and moved any relevant results to section 3.6 (pasted below), lines 277-283. Lines 348-357 have been kept in the discussion.

"3.6 Linear model estimates of ice core annual accumulation from synoptic typing

Multiple linear regression models were generated to estimate the variability in DSS ice core snowfall accumulation explained by a linear combination of the annual frequency of synoptic types (Fig. 6). The models explain 30% of the variability in DSS accumulation for the 1979-2015 period (Fig. 6a), 24% for the 1957-2015 period (Fig. 6b), and 17% for the 1948-2015 period (Fig. 6c). Perhaps not surprisingly, over 1900-2015, the variability explained is not significant (p > 0.05), and is not shown. Summaries of the linear model outputs can be found in Appendix C. The models capture the mean variability in DSS accumulation, but fail to represent the larger extremes (e.g. 1961, 1978, 1981, 1999, 2001). These results indicate that 20CRv3, using a linear combination of synoptic types, can represent the combination of synoptic conditions that are associated with precipitation variability at Law Dome after 1948, but struggles to capture anomalously high or low ice core accumulation years, even in the early 2000s."

**L293-301 Discussion.**

We have kept this section (now lines 357-365) in the revised discussion, with some slight rephrasing.

Author responses to referee comment 2 - Anonymous Referee

Comments on Nilssen et al. "Evaluating the Twentieth Century Reanalysis Version 3 with synoptic typing and East Antarctic ice core accumulation" submitted to Climate of the Past

Owing to limited weather records before the satellite era, understanding long-term variability and inter-decadal pattern in synoptic systems over East Antarctica is challenging. This study evaluated the ability of the Twentieth Century Reanalysis project to reproduce the synoptic conditions associated with increased precipitation at Law Dome since 1948, using daily 500 hPa geopotential height anomalies and the annual snowfall accumulation record from the ice core. The results indicate that this Reanalysis can reliably represent the meridional weather conditions of increased precipitation at Law Dome before the satellite era, and thus extends the time span of available materials for analyzing weather conditions for this region.

I appreciate the objective of this paper, and I am interested in the results and conclusions. However, there are still several issues to be clarified in this study. I recommend that this manuscript needs a major revision before published.

Major points:

1. Since the authors used Twentieth Century Reanalysis Version 3 to perform this study, I have a major concern on the reliability of the data. Especially, the data series for this atmospheric reanalysis may have suffered a "jump" at the ice sheet scale at the beginning of the satellite era. The authors should add some works to prove that it is reliable at least on regional scales (or at Law Dome). This is very important for the analyze, as the major results and conclusions are relied on the 20CRv3 data.

Thank you for this comment. The reliability of 20CRv3 is certainly a major issue that needs to be addressed. None of the twentieth century reanalyses use satellite data, but CERA-20C does show a jump in P-E over the Antarctic ice sheet at the beginning of the satellite era, while 20CR shows a similar jump in 1950 (Wang et al., 2020). We state in lines 53-58 that we chose 20CR over ERA-20C for this study, as 20CR has been determined to be less vulnerable to inhomogeneities due to changes in observation density, as well as the different assimilation schemes. We also discuss the spurious trends in twentieth century reanalyses in lines 307-311, including the jump in 20CR P-E over the East Antarctic Ice sheet. We have added "However, evaporation is considered to be a minor contributor to variability at the Law Dome site, and so is unlikely to have a major effect on this study (Roberts et al., 2015)." in lines 311-313. 20CRv3 does still have a significant increase in the number of assimilated observations over time, and therefore significant changes in error and reliability over time. We agree that determining the reliability at specific regions or points is very important, which is why we believe our study, using a well understood and accurately dated ice core record as comparison to reanalysis, contributes to this question.

2. I suggest that the second part of the manuscript should be changed to "Data and Methods", and that its content needs to undergo a substantial reorganization to make it more coherent. For instance, the headings of 2.1 and 2.3 stand for "Data" rather than "Methods", and 2.4 includes too much information that is not relevant to the heading, such as the division of the period, and the title of 2.3 does not emphasize the classification of precipitation events. The authors should try to separate the description of the data and methods, and introduce each section specifically, such as "snowfall accumulation record from the Law Dome DSS; Twentieth Century Reanalysis version 3...".

Thank you for this comment. We have reorganised these sections to improve coherency. Whilst improving this section we also realised we had neglected to include information on the International Surface Pressure Databank in the original submission as well, so we have now included this in the 'Data Used' section 2.1.3.

The new headings are as follows:

2 Data and Methods (line 66)
2.1 Data Used (line 67)
2.1.1 Twentieth Century Reanalysis version 3 (line 68)
2.1.2 Annual snowfall accumulation record from the Law Dome (DSS) ice core (line 83)
2.1.3 International Surface Pressure Databank (line 115)
2.2 Methods (line 119)
2.2.1 Self-organising map inputs and evaluation (line 120)
2.2.2 20CRv3 daily precipitation and synoptic types (line 145)
2.2.3 Classification of precipitation events (line 151)
2.2.4 Division of time period and correlation between annual precipitation and accumulation (line 160)
2.2.5 Regression analysis of snowfall accumulation using synoptic types (line 175)

3. The authors claim that the ice core record shows high accumulation rates and seasonality at Law Dome (L76-78), so I am concerned about the possibility of extracting seasonal climate signals (synoptic systems and accumulation) from the ice core record. This would not only enable assessing the reliability of the 20CR on a timescale with higher frequency, but also contribute to understanding the seasonal variability of synoptic patterns affecting the Law Dome.

Indeed, understanding the seasonal signals and seasonal variability at Law Dome (and by extension, other high snowfall ice core sites) is really the driving purpose of this study. It is worth stating that while we state that Law Dome shows high accumulation rates, we do not state that Law Dome exhibits 'high seasonality' in that accumulation - in fact, the opposite is true, with Law Dome displaying less seasonal variability in its accumulation regime than other comparable sites (e.g. see Vance et al., 2024). Nonetheless, seasonality in accumulation definitely occurs, and we think that being able to unlock some of the seasonal variability signals from ice core records like Law Dome, either in sea salt (e.g. Vance et al., 2013; Udy et al., 2024), stratigraphy (Zhang et al., 2023) or stable water isotopes (e.g. Jackson et al., 2023) would be a hugely powerful tool in understanding how synoptic scale processes have changed in the past. However, disentangling these signals at less than annual scales in ice core records is currently still challenging, and relies on very detailed and precise dating of the annual layers. This study is another step in that direction, because it gives us evidence that we can explore the interplay between synoptic types and changes in the ice core accumulation, chemistry and stratigraphy prior to the satellite era - e.g., we now have some confidence to use 20CR to explore the seasonal cycle of snowfall at Law Dome for over 60 years. In addition, we provide evidence that a combination of synoptic types can explain

much of the variability in the ice core record, which is exciting as far as being able to interpret past variability. This will help us understand over the longer term when snow falls and how episodic it is - a key piece of information in then exploring any chemical or stratigraphic markers associated with seasonal snowfall. We have added some detail to the end of the introduction and in the discussion to ensure our purpose with this study is clear to the reader. Please see lines 63-65, 352-354.

"Using this same region allows comparison between the previous study and this temporally extended study, and underpins current and future ice core studies investigating decadal changes to the synoptic to seasonal scale variability of snowfall accumulation in Antarctica."

"In addition, this study will greatly contribute to our ability to develop and understand proxies of seasonal and episodic accumulation in ice core records (e.g. see Zhang et al. (2023))."

4. Surface ablation rarely occurs over most of the Antarctic ice sheet, so snowfall accumulation is contributed mainly by precipitation. However, Law Dome is located in the Antarctic coastal region. Studies have been done to show that these areas near the coast are threatened by rainfall from extreme events such as atmospheric rivers. I would therefore suggest that you should distinguish the precipitation pattern (rainfall or snowfall, they have almost opposite effects on accumulation) in this study, rather than comparing precipitation directly to the accumulation from ice core record. Or, another approach is to confirm that rainfall-derived melting or snow blowing is not sufficient to have a significant effect on the inter-annual variability of snow accumulation at Law Dome.

We agree that there seems to be an increasing trend of rainfall events in coastal regions of East Antarctica during extreme events such as atmospheric rivers, and these events are increasingly being catalogued not only on the Antarctic Peninsula, but also coastal East Antarctica. However, rainfall events do not occur at Law Dome DSS, and we know this for sure because rainfall on a snowpack leaves a very clear signal of frozen liquid water that has percolated into the snowpack, as well as a disrupted oxygen isotope record. Law Dome summit is certainly a coastal location, however its elevation (1,370 metres) means its precipitation type remains only snowfall (and a small fraction of diamond dust/clear sky precipitation), because mean annual temperatures are quite low ( $\sim$ -22 °C). Ensuring non-liquid precipitation only was a key aspect of the original site selection for Law Dome . We have no evidence from our numerous overlapping surface and deep ice core records from Dome Summit South (e.g. compiled in Jong et al., 2022) or from stratigraphic studies at the DSS site, that rainfall events have occurred (Zhang et al., 2023), and we know of only one instance of a melt layer (from warmer than average temperatures and high solar radiation) being observed in an ice core drilled on the eastern flank of Law Dome at a lower elevation than DSS (Pers. Comm. David Etheridge 2023). Thus, we don't think it is relevant to differentiate rainfall and snowfall at the DSS site, as no rainfall has historically occurred.

Nonetheless, we have added much more detail to the new section 2.1.2 around the site characteristics at DSS which we believe answers this comment. See lines 87-92.

"Given its relatively high elevation (1,370 metres above sea level), DSS receives precipitation predominantly in the form of snowfall with an insignificant contribution from clear sky precipitation

(diamond dust). We are not aware of any evidence of significant melt or rainfall events at DSS, despite rainfall being possible during extreme warm events along the East Antarctic coastline (Wille et al., 2024b, a). Any rainfall or melt events would appear as clear stratigraphic boundaries in the ice core record, and these have not been observed (van Ommen and Morgan, 1997; Zhang et al., 2023)."

5. Section 3.6 "Linear model estimates of ice core annual accumulation from synoptic typing": The description in this section is too short and the authors should have described it in more detail.

We agree that some more detail was needed, and we have moved some of the detail about the regression models from the discussion into section 3.6 (pasted below), lines 276-284.

**"3.6 Linear model estimates of ice core annual accumulation from synoptic typing**

Multiple linear regression models were generated to estimate the variability in DSS ice core snowfall accumulation explained by a linear combination of the annual frequency of synoptic types (Fig. 6). The models explain 30% of the variability in DSS accumulation for the 1979-2015 period (Fig. 6a), 24% for the 1957-2015 period (Fig. 6b), and 17% for the 1948-2015 period (Fig. 6c). Perhaps not surprisingly, over 1900-2015, the variability explained is not significant (p > 0.05), and is not shown. Summaries of the linear model outputs can be found in Appendix C. The models capture the mean variability in DSS accumulation, but fail to represent the larger extremes (e.g. 1961, 1978, 1981, 1999, 2001). These results indicate that 20CRv3, using a linear combination of synoptic types, can represent the combination of synoptic conditions that are associated with precipitation variability at Law Dome after 1948, but struggles to capture anomalously high or low ice core accumulation years, even in the early 2000s."

6. Although this paper investigated synoptic types on a regional scale, the study relied on ice core records from the Law Dome, so it is inappropriate to show "East Antarctic ice core accumulation" in the title, and I suggest changing it to the "Law Dome". East Antarctica covers a much larger spatial area not studied by this paper, and a single ice core record may not be strongly spatially representative. The authors also mention in the description of L277-279 that the accumulation record will not appear in the Law Dome DSS when the location of the blocking is slightly offset. Therefore, much of this study is not actually representative of East Antarctica.

While it is true that we use only one ice core record to 'ground truth' our findings, we disagree that our title is inappropriate, since we are using the reanalysis to evaluate whether we can look at regional synoptic scale variability with relevance to the Indian Ocean section of East Antarctica. The synoptic variability, and the ability for 20CR to discern precipitation types (e.g. high and extreme precipitation) is the important finding, as this will allow us and others to have confidence in using a reanalysis that is longer than the satellite era for the East Antarctic region. The use of Law Dome is more to check whether the ice core accumulation record is also representative, and then it is used as an example of the application of this study. We think the results from our study are more likely to be utilised for the longer,

regional synoptic typing dataset we have produced, and this implies regional rather than local applicability. We propose a compromise whereby we change the title to:

"Evaluating the Twentieth Century Reanalysis Version 3 with synoptic typing and an East Antarctic ice core accumulation record."

Regarding the location of blocking, we are unclear what the reviewer means here. It is true that the specific location of a block will change the precipitation that results at a specific location (e.g., the moisture transport may be enhanced or blocked at DSS), however we strongly disagree that this means the study is not representative of East Antarctica. The east-west offset referred to in the manuscript is to do with the within-type variability of Synoptic type 3 (lines 340-344), which likely influences the relationship to one specific location (i.e. DSS) but is still representative of the regional signal - supporting the use of East Antarctica in the title.

In contrast, this study extends on a significant body of prior work that shows that the frequency, intensity and location of blocking in the southern Indian Ocean is crucial to understanding precipitation variability across East Antarctica. Please see papers (many of which are referenced in this study) by Jonathan Wille, Benjamin Pohl, Danielle Udy, Mike Pook and Rob Massom for more information.

7. There are some technical corrections in the manuscript, such as the lack of a uniform format for the minus sign "-". In section 3.1, authors sometimes label p<..., sometimes labeled p=.... The authors need to recheck and re-edit them.

We have checked these and made sure they are consistent. For the trends in section 3.1 (lines 187-190) and 3.5 (lines 231-232), we have replaced the p-values with 95% confidence intervals.

**Minor points:**

**1. Please check the units of potential height in the Figures.**

We have checked this, and these are correct as submitted. Figure 1 shows the mean 500 hPa geopotential height in metres, while Figure 3 shows the 500 hPa geopotential height anomaly in metres. The actual height vs the anomaly may have caused some confusion.

**2. L35: please add the references, such as Zhang et al., 2018; Wang et al., 2020 (which has been presented in the references**

We have added the following references, line 40:

Schneider, D. P. and Fogt, R. L.: Artifacts in Century-Length Atmospheric and Coupled Reanalyses Over Antarctica Due To Historical Data Availability, Geophysical Research Letters, 45, 964–973, https://doi.org/10.1002/2017GL076226, \_eprint: https://onlinelibrary.wiley.com/doi/pdf/10.1002/2017GL076226, 2018.

Wang, Y., Hou, S., Ding, M., and Sun, W.: On the performance of twentieth century reanalysis products for Antarctic snow accumulation, Climate Dynamics, 54, 435–455, https://doi.org/10.1007/s00382-019-05008-4, 2020.

Zhang, Y., Wang, Y., Huai, B., Ding, M., Sun, W.: Skill of the two 20th century reanalyses in representing Antarctic near-surface air temperature, Int J Climatol, 38, 4225–4238, https://doi.org/10.1002/joc.5563, 2018.

3. L112: "The 90th and 99th percentile of 20CRv3 daily precipitation at Law Dome was calculated". How was the base period for defining extreme events chosen? Extreme precipitation calculated based on percentile thresholds will be very dependent on the selection of base period.

Thank you for picking this up, this is an important point. The base period was chosen to be 1900-2015, which we will add to the text (line 153). The 90th and 99th percentiles of daily precipitation would have been higher if we had chosen a different base period (eg 1950-2015 or 1979-2015), which would have reduced the amount of annual 20CRv3 precipitation from high and extreme days, and reduced the total number of high and extreme days. Either way, we would still see the increase of high and extreme precipitation days from around 1950. We have noted this in the results (lines 263-267)

"Note that the high/extreme percentiles are calculated based on the 1900-2015 time period for consistency across the study period. While the threshold values increase when calculated over more recent time periods (1950-2015 and 1979-2015), the periods of increased frequency of high/extreme precipitation days around 1912, 1930s and post 1948 remains consistent. Additionally, the 1900-2015 percentile threshold is a similar magnitude to threshold calculated over the satellite era using RACMO2 (Turner et al., 2019)."

**4. Figure 2 and Section 3.1: Please plot the linear trend of the two data series in Figure 2, respectively.**

We would prefer not to do this, because we think that adding trend lines to Figure 2 would be somewhat misleading and potentially cause some confusion about the purpose of the study. Figure 2 is to demonstrate the agreement (or lack thereof, in the first half of the 20th century) between the total annual 20CRv3 precipitation at Law Dome, and the annual snowfall accumulation at Law Dome, along with the change in 20CRv3 precipitation attributed to high/extreme precipitation days at specific points through the 20th century. The overall trends in the two data series are discussed in lines 187-190 and shown in Table B1, but the point of this figure is that linear trends would be an insufficient way to examine or make inferences about the variability through time, as the figure shows not only step changes at different points (e.g. 1948 and 1958) but also trend changes (e.g. 20th century Law Dome precipitation compared to satellite era Law Dome accumulation). These changes are discussed more comprehensively in sections 3.2 (lines 195-199), 3.5 (lines 268-271) and 4.1 (lines 301-307).

"Total annual precipitation from high precipitation days (90-99th percentile) increases from the late 1940s, with a further increase from the mid-late 1950s (Fig. 2). Precipitation that can be attributed to extreme precipitation days (over 99th percentile) increases from the mid-late 1950s, and a further increase from around 1980. Between 1900 and 1956, 9.4% of total precipitation came from high precipitation days, and 0.4% from extreme, and for 1957-2015, 45.7% of total precipitation came from high precipitation days, and 12.9% from extreme."

"The number of zero, high and extreme precipitation days increases over time, along with the number of assimilated observations into ISPDv4.7 in the region 50°-75°S, 40°-180°E (Fig. 4a, b). The 20CRv3 displays a reduced ability to generate high precipitation days before the late 1940s, as well as zero and extreme precipitation days before the late 1950s. Days with zero precipitation increases from the late 1950s, with a continued positive trend until the 2000s."

"Prior to these key observations, variability in annual precipitation derived from the 20CRv3 is low compared to the later period, and does not align with variability in the Law Dome DSS ice core annual snowfall accumulation record (Fig. 2, Table 1). The increase in the number of high/extreme precipitation days explains the apparent positive trend in the 20CRv3

annual precipitation over the period 1900-2015 (Table B1). This positive trend is likely an artefact of increased observations in ISPDv4.7 as the Law Dome DSS annual snowfall accumulation record does not indicate any trends over this time period (Table B1). This spurious trend in the 20CRv3 precipitation cautions the use of trend analysis alone to examine variability over the 20th century. "

5. L177-179: What are the quantitative standards for dividing the weather types? Type 2 also seems to dominate by meridional, despite the blocking high not landing on the ice sheet.

Segregating or dividing the weather types by dominant atmospheric pattern is a subjective process when using this type of analysis, in this and other studies. This means that different interpretations are possible. However, while type 2 does appear more meridional than the other mixed types, we classified it as mixed because the geopotential height anomalies are much weaker than those that are observable in the types we classified as meridional (1, 3, and 4). For this study, we are interested in meridional transport of moisture to East Antarctic region, which requires the block to interact with Antarctic coastline. As the reviewer mentions, the block does not extend to the ice sheet, and we know this is critical in changing the moisture transport regime to the ice sheet (e.g. Jackson et al., 2023, Pohl et al., 2021), so we think that type 2 should remain a mixed rather than meridional type.

We have added "favourable for meridional transport of moisture to East Antarctica" to line 223.

6. The discussion of the relationship between annual frequency of synoptic types and DSS accumulation is relevant and needs to be reflected in the Conclusions and Abstract.

Thank you for this comment. We have added discussion of the relationship between the annual frequency of synoptic types and DSS accumulation to the abstract (lines 15-17) and conclusion (lines 368-370)

"Furthermore, we find a linear combination of the annual frequency of select synoptic types explains a significant amount of the variability in Law Dome snowfall accumulation compared to any individual synoptic type alone."

"Our results indicate that 20CRv3 can reliably represent synoptic conditions associated with precipitation variability at Law Dome from the late 1940s, and that this variability can be at least partially explained using a linear combination of the annual frequency of synoptic types."

References: Zhang Y, Wang Y, Huai B, Ding M, Sun W. Skill of the two 20th century reanalyses in representing Antarctic near-surface air temperature. Int J Climatol. 2018; 38:4225–4238. https://doi.org/10.1002/joc.5563

References:

Jackson, S. L., Vance, T. R., Crockart, C., Moy, A., Plummer, C., and Abram, N. J.: Climatology of the Mount Brown South ice core site in East Antarctica: implications for the interpretation of a water isotope record, Climate of the Past, 19, 1653–1675, https://doi.org/10.5194/cp-19-1653-2023, publisher: Copernicus GmbH, 2023.

Jong, L. M., Plummer, C. T., Roberts, J. L., Moy, A. D., Curran, M. A. J., Vance, T. R., Pedro, J. B., Long, C. A., Nation, M., Mayewski, P. A., and van Ommen, T. D.: 2000 years of annual ice core data from Law Dome, East Antarctica, Earth System Science Data, 14, 3313–3328, https://doi.org/10.5194/essd-14-3313-2022, publisher: Copernicus GmbH, 2022.

Mcmorrow, A., Ommen, T. D. V., Morgan, V., and Curran, M. A. J.: Ultra-high-resolution seasonality of trace-ion species and oxygen isotope ratios in Antarctic firn over four annual cycles, Annals of Glaciology, 39, 34–40, https://doi.org/10.3189/172756404781814609, 2004.

McMorrow, A. J., Curran, M. A. J., Ommen, T. D. V., Morgan, V., Pook, M. J., and Allison, I.: Intercomparison of firn core and meteorological data, Antarctic Science, 13, 329–337, https://doi.org/10.1017/S0954102001000463, 2001.

Morgan, V. I., Wookey, C.W, Li, J., van Ommen, T.D., Skinner, W., and Fitzpatrick, M.F: Site information and initial results from deep ice drilling on Law Dome, Antarctica, J. Glaciol., 43, 3–10, doi:10.3189/S0022143000002768, 1997.

Pohl, B., Favier, V., Wille, J., Udy, D. G., Vance, T. R., Pergaud, J., Dutrievoz, N., Blanchet, J., Kittel, C., Amory, C., Krinner, G., and Codron, F.: Relationship Between Weather Regimes and Atmospheric Rivers in East Antarctica, Journal of Geophysical Research: Atmospheres, 126, e2021JD035 294, https://doi.org/10.1029/2021JD035294, \_eprint: https://onlinelibrary.wiley.com/doi/pdf/10.1029/2021JD035294, 2021. Roberts, J., Plummer, C., Vance, T., van Ommen, T., Moy, A., Poynter, S., Treverrow, A., Curran, M., and George, S.: A 2000-year annual record of snow accumulation rates for Law Dome, East Antarctica, Climate of the Past, 11, 697–707, https://doi.org/10.5194/cp-11-697- 2015, publisher: Copernicus GmbH, 2015.

Udy, D.G., Vance, T.R., Kiem, A.S., Holbrook, N.J, and Abram, N.: Australia's 2019/20 Black Summer fire weather exceptionally rare over the last 2000 years, Commun Earth Environ, 5, 317, https://doi.org/10.1038/s43247-024-01470-z, 2024.

Vance, T. R., Ommen, T. D. v., Curran, M. A. J., Plummer, C. T., and Moy, A. D.: A Millennial Proxy Record of ENSO and Eastern Australian Rainfall from the Law Dome Ice Core, East Antarctica, Journal of Climate, 26, 710–725, https://doi.org/10.1175/JCLI-D-12-00003.1, publisher: American Meteorological Society Section: Journal of Climate, 2013.

Vance, T. R., Abram, N. J., Criscitiello, A. S., Crockart, C. K., DeCampo, A., Favier, V., Gkinis, V., Harlan, M., Jackson, S. L., Kjær, H. A., Long, C. A., Nation, M. K., Plummer, C. T., Segato, D., Spolaor, A., and Vallelonga, P. T.: An annually resolved chronology for the Mount Brown South ice cores, East Antarctica, Climate of the Past, 20, 969–990, https://doi.org/10.5194/cp-20-969-2024, publisher: Copernicus GmbH, 2024.

Wang, Y., Hou, S., Ding, M., and Sun, W.: On the performance of twentieth century reanalysis products for Antarctic snow accumulation, Climate Dynamics, 54, 435–455, https://doi.org/10.1007/s00382-019-05008-4, 2020.

Zhang, L., Vance, T. R., Fraser, A. D., Jong, L. M., Thompson, S. S., Criscitiello, A. S., and Abram, N. J.: Identifying atmospheric processes favouring the formation of bubble-free layers in the Law Dome ice core, East Antarctica, The Cryosphere, 17, 5155–5173, https://doi.org/10.5194/tc-17-5155-2023, publisher: Copernicus GmbH, 2023.